# A novel LRR receptor-like kinase BRAK reciprocally phosphorylates PSKR1 to enhance growth and defense in tomato

Shuting Ding[1,2,6], Shuxian Feng[1,6], Shibo Zhou[1,6], Zhengran Zhao[1], Xiao Liang[1], Jiao Wang[1], Ruishuang Fu [1], Rui Deng [1], Tao Zhang[3], Shujun Shao[1], Jingquan Yu [1], Christine H Foyer [4] & Kai Shi [1,2,5✉]

## Abstract

**Plants face constant threats from pathogens, leading to growth retardation and crop failure. Cell-surface leucine-rich repeat receptor-like kinases (LRR-RLKs) are crucial for plant growth and defense, but their specific functions, especially to necrotrophic fungal pathogens, are largely unknown. Here, we identified an LRR-RLK (Solyc06g069650) in tomato (*Solanum lycopersicum*) induced by the economically important necrotrophic pathogen *Botrytis cinerea*. Knocking out this LRR-RLK reduced plant growth and increased sensitivity to *B. cinerea*, while its overexpression led to enhanced growth, yield, and resistance. We named this LRR-RLK as BRAK (*B. cinerea* resistance-associated kinase). Yeast two-hybrid screen revealed BRAK interacted with phytosulfokine (PSK) receptor PSKR1. PSK-induced growth and defense responses were impaired in *pskr1*, *brak* single and double mutants, as well as in *PSKR1*-overexpressing plants with silenced *BRAK*. Moreover, BRAK and PSKR1 phosphorylated each other, promoting their interaction as detected by microscale thermophoresis. This reciprocal phosphorylation was crucial for growth and resistance. In summary, we identified BRAK as a novel regulator of seedling growth, fruit yield and defense, offering new possibilities for developing fungal disease-tolerant plants without compromising yield.**

**Keywords** Receptor-like Kinase; *Botrytis cinerea*; Growth and Defense; Phytosulfokine; *Solanum lycopersicum*
**Subject Categories** Microbiology, Virology & Host Pathogen Interaction; Plant Biology

## Introduction

Plants are frequently attacked by various pathogens and herbivores in natural environments. To cope with these continuous threats, plants have developed sophisticated immune systems, that resist pathogen invasion through sensing, signal transduction, and activation of downstream functional gene expression (Ahanger et al, 2017). Due to limited resources obtained from the environment, the activation of plant defense is often accompanied by growth and development inhibition, a phenomenon recognized as the growth-defense trade-off strategy (Brown, 2002; He et al, 2022). These physiological processes are crucial for plant growth and reproduction, particularly in the context of global climate change.

Rapid and accurate detection of pathogens or pathogen-associated signals is a prerequisite for initiating plant defense responses. Numerous pattern recognition receptors (PRRs) on plant cell surfaces detect diverse pathogen-derived elicitors (Schwessinger and Ronald, 2012; Ngou et al, 2022). Among them, receptor-like kinases (RLKs) and receptor-like proteins (RLPs) are particularly essential. RLKs are one of the largest subfamilies of membrane proteins, containing an extracellular domain, a transmembrane region and a cytoplasmic kinase domain. Extracellular signals are recognized by RLKs and are typically transduced via phosphorylation (Lin et al, 2014; He and Wu, 2016). Among RLKs, the leucine-rich repeat RLKs (LRR-RLKs) with a leucine-rich repeat extracellular domain are one of the largest and most widely studied classes (Ma et al, 2022). There are 223 LRR-RLK genes in the Arabidopsis genome, 309 in the rice genome, and 234 in the tomato genome (Antolin-Llovera et al, 2014; Wu et al, 2016; Liu et al, 2017). Furthermore, LRR-RLKs also form complexes with each other to carry out their functions (Wang et al, 2018; Ma et al, 2022). For example, the well-known LRR-RLK, FLS2 recognizes bacterial flagellin and forms a complex with BAK1 to initiate immunity to *Pseudomonas syringae* pv. *tomato* (*Pst*) strain DC3000 (Chinchilla et al, 2007). FIR (FLS2-interacting receptor) promotes BAK1-FLS2 complex formation to trigger the immune response (Smakowska-Luzan et al, 2018).

LRR-RLKs are involved not only in resistance but also in plant growth. For instance, BRI1 (BR-INSENSITIVE 1) perceives brassinosteroid (BR), which regulates cell elongation, photomorphogenesis, and other aspects of plant growth processes (Guo et al, 2013). Mutation of barley BRI increased immunity to the hemibiotrophic pathogens *Magnaporthe oryzae* and *Fusarium culmorum* while reducing the plant growth rate (Goddard et al, 2014). CLAVATAs (CLVs) play a crucial role in regulating the

[1]Department of Horticulture, Zhejiang University, 866 Yuhangtang Road, 310058 Hangzhou, China. [2]Hainan Institute, Zhejiang University, Yazhou Bay Science and Technology City, 572025 Sanya, China. [3]College of Horticulture, Henan Agricultural University, 450002 Zhengzhou, China. [4]School of Biosciences, College of Life and Environmental Sciences, University of Birmingham, Edgbaston B15 2TT, UK. [5]Zhejiang Key Laboratory of Horticultural Crop Quality Improvement, 310058 Hangzhou, China. [6]These authors contributed equally: Shuting Ding, Shuxian Feng, Shibo Zhou. ✉E-mail: kaishi@zju.edu.cn

development and maintenance of plant stem cells (Nimchuk et al, 2011), and Arabidopsis *clv1* and *clv2* mutants confer enhanced disease resistance to bacterial wilt caused by *Ralstonia solanacearum* (Hanemian et al, 2016). The T-DNA insertion mutant *rlk902* displayed a reduced meristem size and root length, and showed enhanced resistance to downy mildew in Arabidopsis (Ten et al, 2011). It seems likely that the majority of LRR-RLKs discovered so far play opposing roles in terms of growth and defense. Moreover, previous research has predominantly focused on the immune function of LRR-RLKs in response to bacterial and oomycete pathogens (Wu and Zhou, 2013; Peng et al, 2015; Hu et al, 2018; Zhang et al, 2022), with few studies investigating the role of LRR-RLKs in plant defense against fungal pathogens. Throughout agricultural history, phytopathogenic fungi have posed devastating threats, leading to an annual loss of approximately 10% of agricultural production, and these losses are expected to increase with the growing consequences of climate change (Lo Presti et al, 2015). Therefore, there is an urgent need to identify LRR-RLKs that can enhance resistance to fungal pathogens while simultaneously not compromising growth and, ideally, promoting growth.

Tomato (*Solanum lycopersicum* L.) is one of the most widely consumed and commercially valuable vegetables worldwide, but its production is frequently threatened by a range of fungal pathogens. In particular, *Botrytis cinerea*, a necrotrophic fungus of significant scientific and economic concern, causes gray mold disease and severely affects the production of more than 200 host species, including tomato (Dean et al, 2012). *B. cinerea* can infect different stage of crops, from the seedling stage to maturity, and different plant organs, including leaves, flowers, shoots and fruits, resulting in an annual global loss of produce estimated at $10 to $100 billion (Weiberg et al, 2013). No plant shows complete resistance to *B. cinerea*, and chemical methods are the most common means to control its spread. However, the use of fungicides is harmful to both the environment and human health. Worse still, resistance of *B. cinerea* to fungicides develops rapidly (Bi et al, 2023). Therefore, investigations into the roles of LRR-RLKs in tomato would open up new possibilities for breeding tomato cultivars with improved growth and defense against *B. cinerea*. In this study, we identified a novel LRR-RLK (Solyc06g069650) by analyzing the expression patterns of 234 tomato LRR-RLK genes in response to *B. cinerea* infection, and named it *B. cinerea* resistance-associated kinase (*BRAK*). The study showed that BRAK interacted with the PSK peptide receptor PSKR1 to function positively in growth and resistance to *B. cinerea*, and that BRAK's function was related to the PSK signaling pathway through BRAK-PSKR1 reciprocal phosphorylation. This research adds novel insights into the role of LRR-RLK complexes in plant growth and defense, and indicates that BRAK can be an attractive candidate for tomato breeding programs to improve resistance to *B. cinerea* and enhance plant growth.

## Results

### *BRAK* is induced by *B. cinerea* inoculation in tomato leaves

To evaluate the transcriptional response of tomato LRR-RLKs to *B. cinerea* infection, we analyzed the gene expression data from a published transcriptome profile of *B. cinerea*-infected tomato leaves

(Courbier et al, 2021). Among the 234 LRR-RLKs, 110 exhibited differential expression 30 h post-inoculation (hpi) with *B. cinerea*: 39 genes were upregulated, and 71 were downregulated. Notably, all differentially expressed genes in the LRR-RLK subfamily IV were upregulated, showing the highest number of upregulated genes among all subfamilies (Fig. 1A; Dataset EV1). We further validated the expression of subfamily IV members by qRT-PCR at 30 hpi with *B. cinerea*, finding that the transcriptional abundance of Solyc06g069650, henceforth named *B. cinerea* resistance-associated kinase (*BRAK*), was the highest, corroborating the transcriptome data (Fig. 1B).

The subcellular localization of BRAK was determined by transfecting tomato mesophyll protoplasts with BRAK fused to GFP, which revealed localization at the plasma membrane (Fig. 1C). The protein structure of BRAK predicted by SMART, Pfam and TMHMM V2.0, included extracellular leucine-rich domains, a transmembrane domain, and an intracellular kinase domain, consistent with a typical LRR-RLK structure (Appendix Fig. S1A). Furthermore, BRAK expression was detected to be throughout the plant, including roots, stems, leaves, flowers, fruits, and seeds (Appendix Fig. S1B).

### BRAK promotes tomato seedling growth, fruit yield and defense against *B. cinerea*

To elucidate the biological function of BRAK, we generated tomato *BRAK* knock-out and overexpression lines using CRISPR/Cas9-mediated gene editing and transgenic techniques, respectively. Two homozygous gene-edited lines *brak*#4 and *brak*#5 (with 2 and 1 bp deletions causing frame change with stop codons at 268 and 307 bp, respectively), and *BRAK*-overexpressing lines, OE-*BRAK*#3 and OE-*BRAK*#6, were isolated for further experiments (Fig. 2A,B).

The *brak* mutants exhibited approximately 10% shorter plant height than WT plants at 4 weeks post-sowing and had shorter root lengths than WT plants 3 days after transfer to 1/2 MS medium. In contrast, OE-*BRAK* plants showed increased plant height and root length (Fig. 2C–F). Moreover, fruit size and number per plant were significantly higher in OE-*BRAK* plants compared to WT, resulting in total fruit yield increases of 12.47% and 14.56% in OE-*BRAK*#3 and OE-*BRAK*#6, respectively, while yield was reduced in both *brak* mutant lines (Fig. 2G–K).

To study the function of BRAK in defense against *B. cinerea*, WT, *brak* mutants, and OE-*BRAK* plants were inoculated with *B. cinerea*. The *brak* mutants exhibited significantly reduced resistance to *B. cinerea*, demonstrated by increased leaf lesions, elevated *B. cinerea actin* mRNA accumulation, and decreased photosystem II quantum yield (ΦPSII). Conversely, OE-*BRAK* plants showed enhanced resistance to *B. cinerea* compared to WT, as reflected by these parameters (Fig. 2L–O). These findings reveal that BRAK enhances plant immunity to *B. cinerea* in tomato.

### BRAK interacts with PSKR1 and responds to PSK

To uncover the mechanisms underlying BRAK-mediated plant growth and *B. cinerea* resistance, a yeast two-hybrid (Y2H) screen was conducted using the juxtamembrane and kinase domains of BRAK (BRAKJK) as bait. Transformants expressing interacting pairs of pGBKT7-BRAKJK and AD library proteins in AH109 yeast cells were screened on SD medium lacking Trp, Leu, Ade, and His.

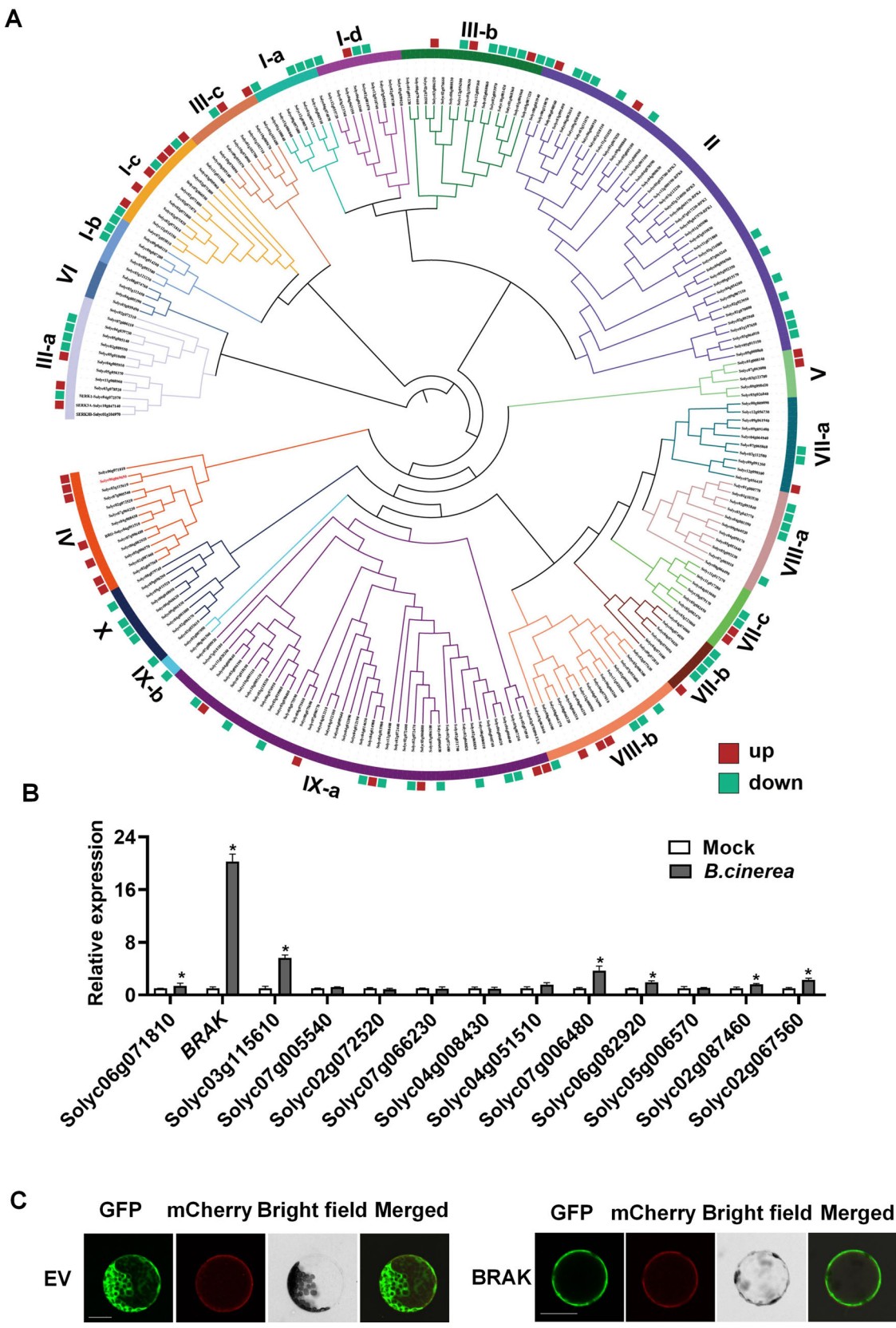

◀ **Figure 1.  Expression of LRR-RLK family genes after *B. cinerea* inoculation.**

(**A**) The transcript abundance of tomato LRR-RLK family genes in leaves 30 h post *B. cinerea* inoculation. (**B**) The transcript abundance of tomato LRR-RLK subfamily IV genes in leaves 30 h post *B. cinerea* inoculation. The gene expression level under mock condition was defined as 1. (**C**) Subcellular localization of BRAK. The construct expressing BRAK-GFP and the plasma membrane-localized protein FLS2-mCherry were co-transfected in tomato mesophyll protoplasts. Empty vector (EV) expressing GFP was used as a negative control. Scale bar, 25 μm. Data in (**B**) is presented as mean values ± SD, *n* = 3 independent pooled samples with each sample being from two plants in (**B**). The asterisks depict statistically significant differences, as analyzed by one-way ANOVA with Tukey's HSD post hoc test (*P* < 0.05). Experiments were repeated three times with similar results. Source data are available online for this figure.

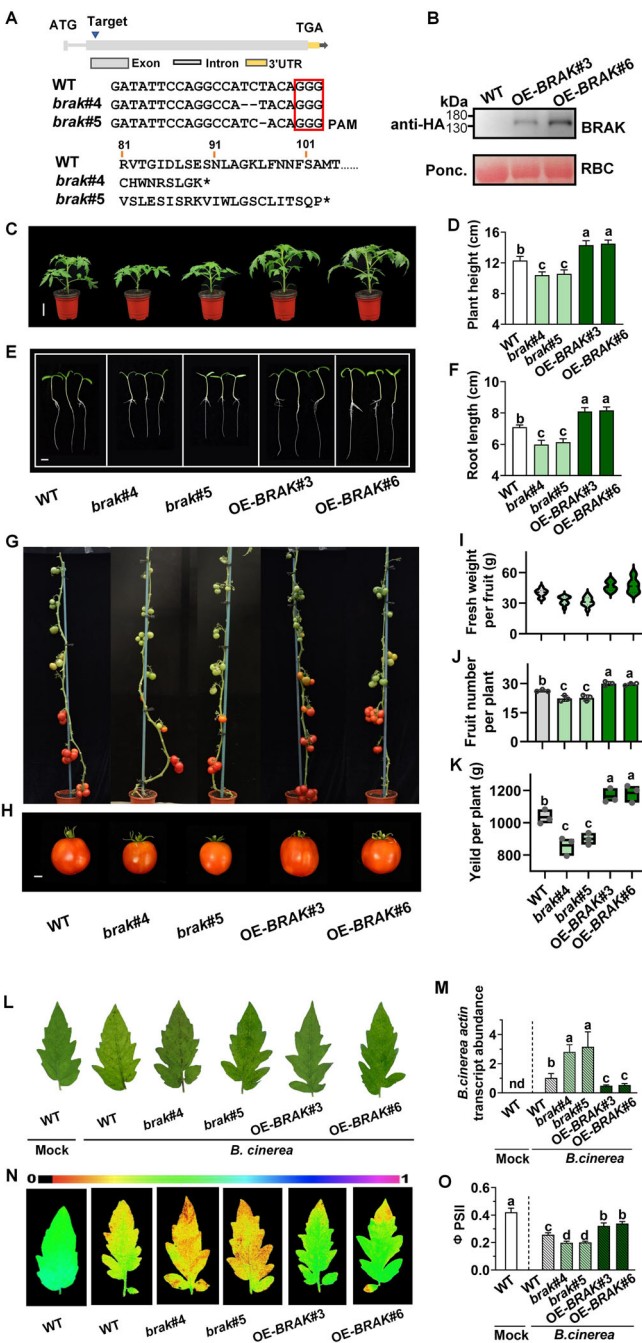

**Figure 2.  Functions of *BRAK* in tomato growth and defense against *B. cinerea*.**

(**A**) Schematic illustration of the sgRNA target site and protein sequence of wild-type (WT) BRAK and the allele from CRISPR/Cas9 T2 mutant lines (*brak*#4, 5). Dash presents the deletion sequences, the red box presents protospacer-adjacent motif (PAM) sequences, the blue arrow indicates the sgRNA target site and the asterisk presents the termination of the protein sequence. (**B**) Detection of the BRAK-HA fusion protein in *BRAK*-overexpressing lines (OE-*BRAK*#3, 6) by western blot using an anti-HA antibody and the large subunit of Rubisco (RBC, as a loading control). (**C–F**) Phenotype and index of plant seedling growth. The representative plant image (**C**) and plant height data (**D**) of 4-week-old WT, *brak* and OE-*BRAK* were shown. The representative image (**E**) and root length data (**F**) of WT and each line grown on 1/2 MS medium for 3 days. Bar, 5 cm (**C**), 1 cm (**E**). (**G, H**) Representative plant fruit phenotype in the reproductive stage. Bar, 1 cm (**B**). (**I–K**) Fresh weight per fruit (**I**), fruit number per plant (**J**) and yield (**K**). (**L–O**) Phenotype and index of plants inoculated with *B. cinerea*. (**L**) Disease symptoms photographed at 3 days post inoculation (dpi). (**M**) Relative *B. cinerea actin* transcript abundance in infected leaves at 1 dpi. Representative chlorophyll fluorescence imaging (**N**) and quantification (**O**) of ΦPSII at 3 dpi. Data in (**D, F, I, J, K, M, O**) are presented as mean values ± SD, *n* = 8 different plants in (**D**), *n* = 8 different tomato seedings in (**F**), *n* = 10 different tomato fruits in (**I**), *n* = 3 individual plants in (**J**), *n* = individual 3 plants in (**K**), *n* = 3 independent pooled samples with each sample being from two plants in (**M**), *n* = 8 leaves from different plants in (**O**). For boxplots, the center line in the box indicates median, dots represent data, limits represent upper and lower quartiles. Different letters depict statistically significant differences, as analyzed by one-way ANOVA with Tukey's HSD post hoc test (*P* < 0.05). Source data are available online for this figure.

Plasmids from positive clones were isolated and sequenced, identifying 53 candidate BRAK-interacting proteins (Dataset EV2). Among them, two LRR-RLKs were identified, one being Solyc04g064940 which belongs to VII-a, showing no significant response to *B. cinerea*. The other one was PSKR1 (Solyc01g008140), significantly induced by *B. cinerea*, was confirmed to interact with BRAK by Y2H (Fig. 3A).

PSKR1, has been reported as the receptor of PSK, an endogenous disulfated pentapeptide acting as a damage-associated molecular patterns (DAMP) in immune responses. In *Arabidopsis*, PSKR1 is associated with root development and resistance to fungal pathogens such as *Alternaria brassicicola*, *Sclerotinia sclerotiorum*, and the bacterial pathogen *Pst* DC3000 (Loivamäki et al, 2010; Igarashi et al, 2012; Mosher et al, 2013; Rodiuc et al, 2016; Kaufmann et al, 2021). In tomato, PSKR1 has been reported to play a role in defense against *B. cinerea* (Zhang et al, 2018; Hu et al, 2023). The interaction between BRAK and PSKR1 was further confirmed via co-immunoprecipitation (co-IP) assays by expressing differently tagged proteins in *Nicotiana benthamiana*, showing that BRAK associates with PSKR1 through their juxtamembrane and kinase domains (JK) (Fig. 3B). Notably, PSK treatment enhanced the interaction between BRAK and

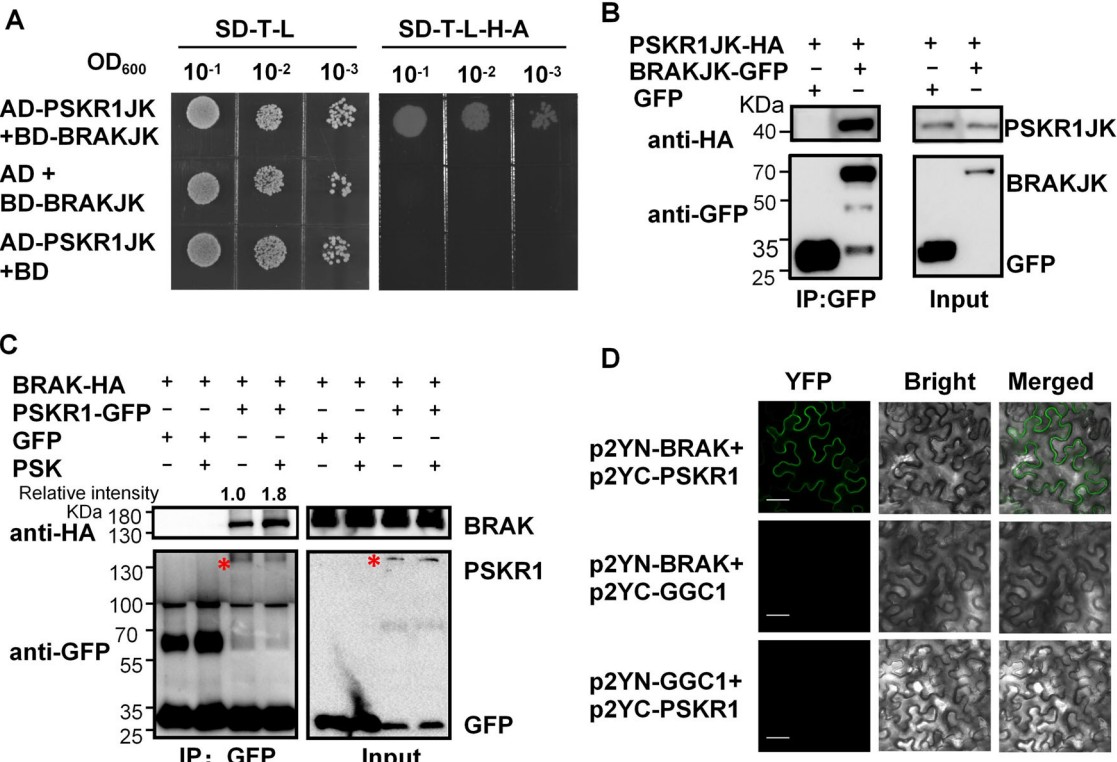

**Figure 3.  BRAK interacts with PSKR1.**

(**A**) Y2H assay. The interactions between the GAL4 DNA activation domain (AD) fusion of PSKR1JK, and GAL4 DNA-binding domain (BD) fusion of BRAKJK were detected. The transformed yeast cells were grown on selective media lacking Trp (T), His (H), Ade (A), and Leu (L). JK presents the inner juxtamembrane domain and the protein kinase domain. (**B**, **C**) Co-immunoprecipitation assays. (**B**) PSKR1JK-HA were co-expressed with BRAKJK-GFP or GFP in *N. benthamiana*. (**C**) BRAK-HA was co-expressed with PSKR1-GFP or GFP in *N. benthamiana*. Two days later, PSK (10 µM) or H₂O were treated two hours before leaves sampling. Proteins were subjected to immunoprecipitation (IP) with GFP trap beads followed by immunoblot with anti-HA or anti-GFP antibodies. Asterisks indicate the specific bands of PSKR1-GFP. (**D**) BiFC assays. p2YN-BRAK and p2YC-PSKR1 were co-expressed in *N. benthamiana* leaves. GGC1 was used as the negative control. The YFP fluorescence was detected using a confocal microscope 2 days later. Bar, 50 µm. The above experiments were performed twice with similar results. Source data are available online for this figure.

PSKR1 (Fig. 3C). The BRAK-PSKR1 complex was localized at the cell membrane according to the BiFC assay (Fig. 3D).

## BRAK-regulated growth and defense is related to the PSK signaling pathway

Given that both BRAK and PSKR1 are essential for plant growth and defense against *B. cinerea*, we investigated their relationship within the PSK signaling pathway. We generated *pskr1 brak* double mutants by crossing various genotypes. As shown in Fig. 4A,B, exogenous PSK promoted root length in WT, but had no significant effect on *pskr1*, *brak* single mutants, or *pskr1 brak* double mutants. Although PSK alleviated *B. cinerea*-induced leaf lesions, reductions in ΦPSII, and increases in *B. cinerea* actin mRNA abundance in WT, these effects were absent in *pskr1*, *brak* single mutants, and *pskr1 brak* double mutants (Fig. 4C–F). Furthermore, we silenced the *BRAK* gene in OE-*PSKR1* plants due to the failure to obtain OE-*PSKR1 brak* plants. PSK-induced root elongation in both WT and OE-*PSKR1* was compromised by *BRAK* gene silencing (Fig. 5A,B; Appendix Fig. S2A). Similarly, the enhanced defense conferred by OE-*PSKR1* and exogenous PSK was impaired by *BRAK* silencing (Fig. 5C–F; Appendix Fig. S2B). The fruit number

and total fruit yield were increased in OE-*PSKR1* plants and decreased in *pskr1* mutants (Fig. EV1). These observations suggest that BRAK and PSKR1 function in the same pathway, with BRAK likely acting downstream of PSK to regulate tomato seedling growth, yield, and immunity to *B. cinerea*.

To further understand the role of BRAK in PSK signaling, the transcriptomes of WT and *brak* mutant seedlings after PSK treatment were compared. WT and *brak* (#5) mutants grown on 1/2 MS media were treated with or without 1 µM PSK, and roots were sampled 12 h later. Upon PSK treatment, 752 transcripts were differentially expressed (fold change ≥2, $P < 0.05$) in WT, while only 28 transcripts were differentially expressed in *brak* mutants (Fig. 5G; Dataset EV3). Among these, 749 transcripts were significantly altered in WT compared to *brak* mutants, indicating that the expression of 96.39% of PSK-regulated genes was BRAK-dependent (Fig. 5H). These findings confirm the essential role of BRAK in PSK signaling.

## The role of kinase activity in the BRAK/ PSKR1 association

To elucidate the detailed mechanisms of BRAK/PSKR1 association in plant growth and defense, we examined the kinase activity of

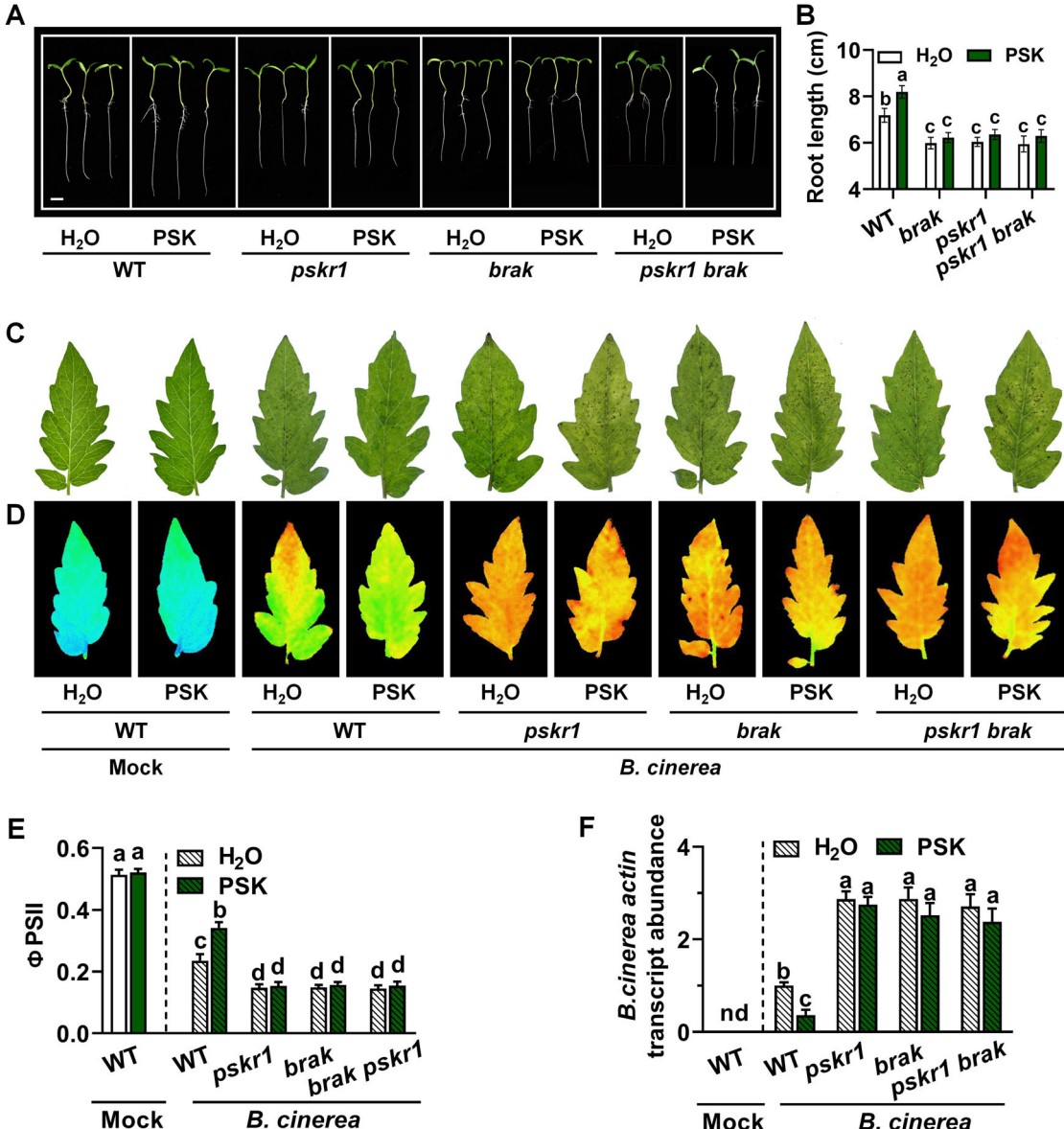

**Figure 4. BRAK is involved in PSK-induced tomato growth and defense against *B. cinerea*.**

(A, B) Phenotype and index of plant seedling growth. The representative image (A) and root lengths data (B) of tomato WT, *pskr1*, *brak* single mutants and *pskr1 brak* double mutants grown on 1/2 MS medium with or without 1 μM PSK for 3 days. Bar = 1 cm. (C–F) Phenotype and index of plants infected by *B. cinerea*. Representative leave image (C), chlorophyll fluorescence of ΦPSII (D), and quantification of ΦPSII (E) were collected at 3 dpi. (F) Relative *B. cinerea actin* transcript abundance in leaves at 1 dpi. Leaves of 4-week-old tomato were treated with 10 μM PSK or H₂O 12 h before *B. cinerea* inoculation. Data in (B, E, F) are presented as mean values ± SD, *n* = 8 different tomato seedings in (B), *n* = 8 leaves from different plants in (E), *n* = 3 independent pooled samples with each sample being from two plants in (F). Different letters depict statistically significant differences, as analyzed by one-way ANOVA with Tukey's HSD post hoc test (*P* < 0.05). These experiments were performed three times with similar results. Source data are available online for this figure.

BRAK and PSKR1. BRAK-GFP and PSKR1-HA were co-expressed in *N. benthamiana* leaves. Based on immunoblotting with anti-phospho-serine/threonine (anti-pSer/Thr) and anti-phospho-tyrosine (anti-pTyr) antibodies, we observed increased phosphorylation levels of BRAK upon PSK treatment (Fig. 6A). To identify the PSKR1-mediated phosphorylation sites on BRAK, the kinase-inactive mutant GST-BRAKJK^KM, which bears a lysine (K)-to-glutamate (E) substitution in the ATP-binding site (K830E) (Fig. EV2A), was incubated with His-PSKR1JK in vitro. Two sites,

threonine 1027 (T1027) and tyrosine 1048 (Y1048) on the BRAKJK protein were identified to be potentially phosphorylated by PSKR1 with LC-MS/MS analysis (Figs. 6B and EV2B). Next, we created BRAKJK^T1024A and BRAKJK^Y1048F variants, by substituting these residues with non-phosphorylatable alanine (A) and phenylalanine (F), respectively. Microscale thermophoresis (MST) measurements showed that the dissociation constant (Kd) of PSKR1JK with BRAKJK was lower than that with BRAKJK^T1024A or BRAKJK^Y1048F, indicating that PSKR1 and BRAK interaction was enhanced

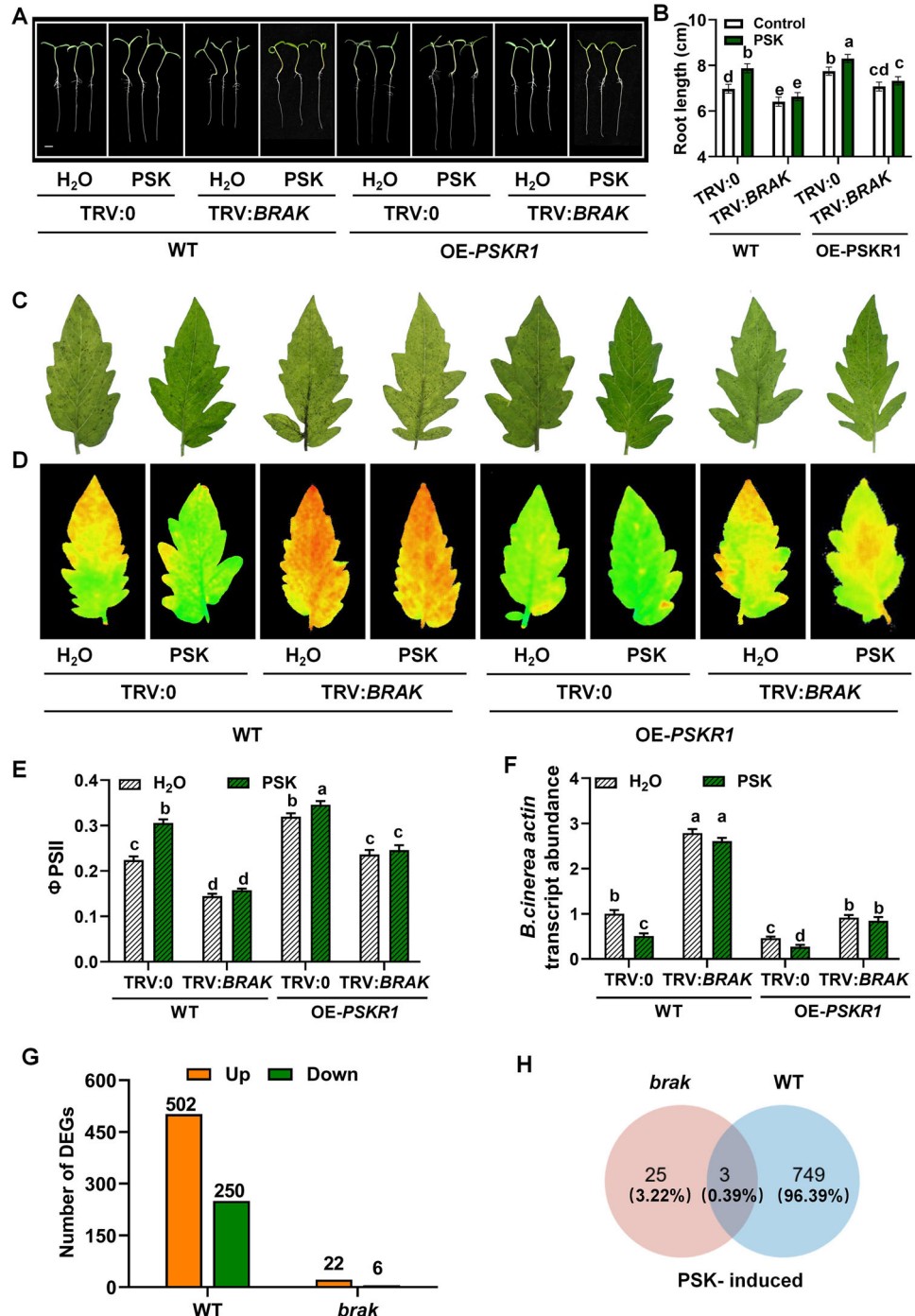

**Figure 5. BRAK is related to PSK signaling.**

(A, B) Phenotype and index of plant seedling growth. The representative image (A) and root length (B) data of tomato VIGS plants growing on 1/2 MS medium for 3 days with or without PSK treatment. Bar = 1 cm. (C–F) Phenotype and index of plants inoculated with *B. cinerea*. (C) Disease symptoms were photographed at 3 dpi. (D) Relative *B. cinerea actin* transcript abundance in leaves at 1 dpi. Representative chlorophyll fluorescence imaging (E) and quantification (F) of ΦPSII at 3 dpi. Leaves of 4-week-old tomato were treated with 10 μM PSK or H₂O 12 h before *B. cinerea* inoculation. (G) Numbers of differentially PSK-changed genes (PSK vs H₂O, fold change ≥2 and *P* < 0.05) in WT and *brak*#5 mutants. Root samples were collected 12 h after PSK-treatment. (H) Venn diagram showing the number of PSK-induced genes (fold change ≥2 and *P* < 0.05) in WT and *brak* mutants. Data in (B, E, F) are represented as mean values ± SD, *n* = 8 different tomato seedings in (B), *n* = 8 leaves from different plants in (E), *n* = 3 independent pooled samples with each sample being from two plants in (F). Different letters depict statistically significant differences, as analyzed by one-way ANOVA with Tukey's HSD post hoc test (*P* < 0.05). Experiments in (A–F) were performed three times with similar results. Source data are available online for this figure.

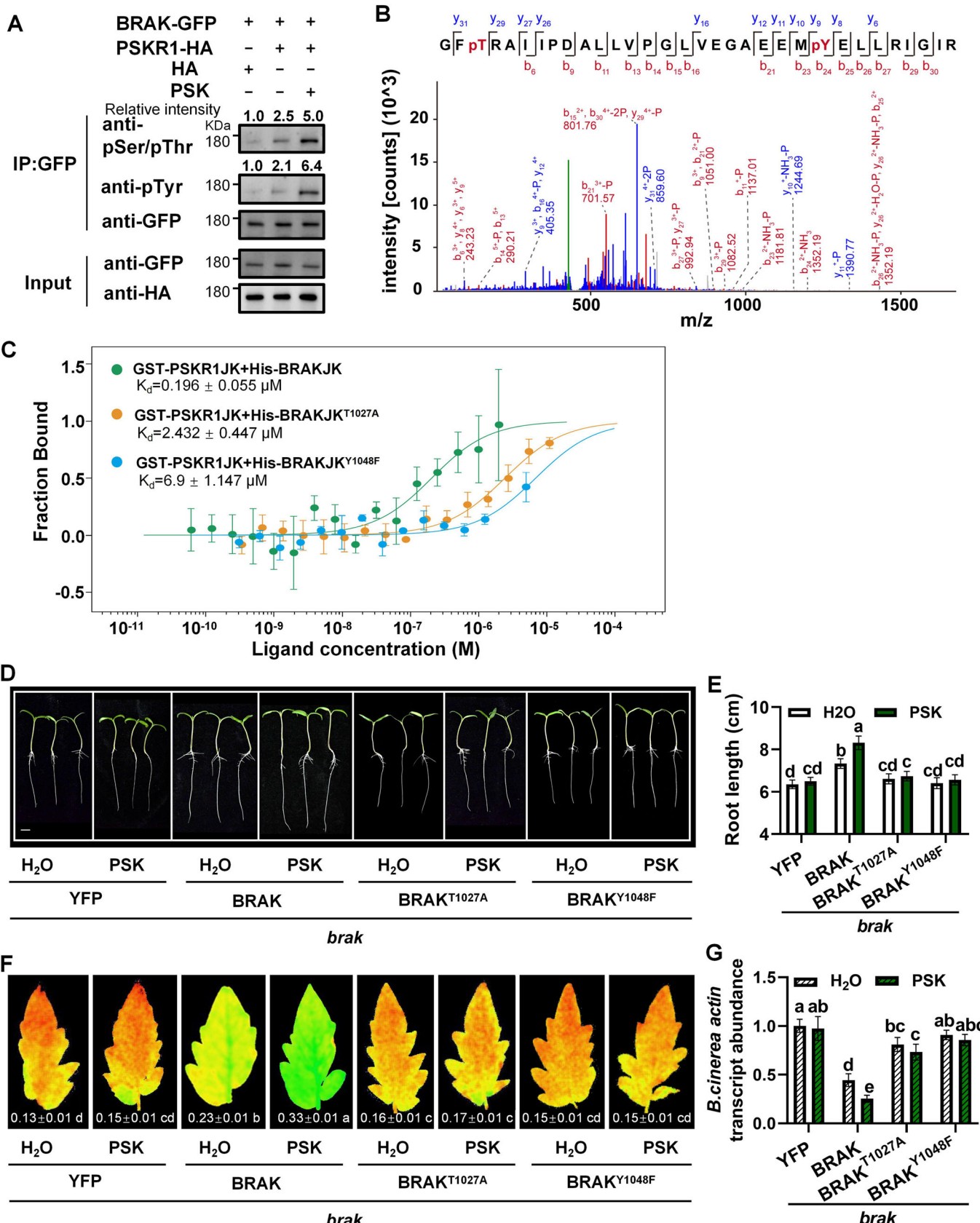

© The Author(s)

**Figure 6.   PSKR1 phosphorylates BRAK at T1027 and Y1048 to regulate PSK-induced tomato growth and defense against *B. cinerea*.**

(**A**) PSK induces the phosphorylation of BRAK by PSKR1 in vivo. BRAK-GFP was co-expressed with PSKR1-HA (empty vector as a negative control) in *N. benthamiana* leaf. And 10 μM PSK or $H_2O$ was injected into the leaf before sampling. The phosphorylated BRAK protein was precipitated by GFP-Trap, and then immunoblotted with anti-pSer/Thr and anti-pTyr antibodies. (**B**) In vitro mass spectrometric analysis of BRAK phosphorylation site by PSKR1. BRAK T1027 site and Y1048 site. Threonine and tyrosine residues in the BRAK are highlighted in red. (**C**) In vitro interaction between PSKR1JK and wild-type or phospho-dead BRAKJK revealed using MST. PSKR1-GST was regarded as the target, and the WT or phospho-dead BRAKJK protein was used as the ligand. (**D, E**) Phenotype and index of plant seedling growth. The representative image (**D**) and root lengths data (**E**) of *brak* mutant transiently overexpressed with YFP (as a negative control), wild-type BRAK or phospho-dead BRAK grown on 1/2 MS medium with or without 1 μM PSK for 5 days. Bar = 1 cm. (**F, G**) Phenotype and index of plants infected by *B. cinerea*. Chlorophyll fluorescence of ΦPSII (**F**), and *B. cinerea actin* transcript abundance (**G**) in leaves at 1 dpi. Leaves of 4-week-old tomato were treated with 10 μM PSK or $H_2O$ 12 h before *B. cinerea* inoculation. Data in (**E–G**) are presented as mean values ± SD, $n = 8$ different tomato seedings in (**E**), $n = 8$ leaves from different plants in (**F**), $n = 4$ independent pooled samples with each sample being from two plants in (**G**). Different letters depict statistically significant differences, as analyzed by one-way ANOVA with Tukey's HSD post hoc test ($P < 0.05$). Experiments in (**A–C**) were performed twice, and experiments in (**D–G**) were repeated three times with similar results. Source data are available online for this figure.

through phosphorylation (Fig. 6C). To gain further insight into the dynamics of PSKR1-BRAK interaction, we assessed the effect of their interaction on autophosphorylation. Purified BRAKJK$^{KM}$ or PSKR1JK$^{KM}$ proteins was co-incubated with wild-type PSKR1JK or BRAKJK, respectively, revealing that their autophosphorylation levels were unaffected by their interaction (Fig. EV2D,E).

To determine the functional relevance of these residues, wild-type BRAK and site-directed mutants were transiently overexpressed in the tomato *brak* background. And YFP was used as a negative control (Appendix Fig. S3A,B). Overexpressing wild-type BRAK rescued root length and *B. cinerea* resistance in the *brak* mutant, while BRAK$^{T1027A}$ and BRAK$^{Y1048F}$ mutants did not, highlighting reduced PSK signaling output (Fig. 6D–G).

Since reciprocal phosphorylation occurs between RLKs (Wang et al, 2005; Lu et al, 2010; Li et al, 2019), we explored whether BRAK phosphorylates PSKR1. In vitro and in vivo assays confirmed that BRAK phosphorylates PSKR1 (Figs. 7A and EV2C). LC-MS/MS analysis identified phosphorylation sites at tyrosine 843 (Y843) and threonine 890 (T890) on PSKR1 (Fig. 7B,C). MST analysis demonstrated weakened interaction strength between BRAK and PSKR1$^{Y843F}$ or PSKR1$^{T890A}$ compared to wild-type PSKR1 (Fig. 7D). Similarly, overexpressing wild-type PSKR1 in *pskr1* mutants improved root lengths and *B. cinerea* resistance, while the site-directed mutants PSKR1$^{Y843F}$ and PSKR1$^{T890A}$ did not fully rescue these phenotypes or PSK responses (Fig. 7E–H; Appendix Fig. S3C,D). These findings indicate that PSK enhances mutual phosphorylation of BRAK and PSKR1, crucial for plant growth and defense.

## BRAK-regulated plant defense is associated with the oxidation-reduction process and protein phosphorylation

To further investigate how BRAK regulates plant defense against *B. cinerea*, RNA-seq analysis was performed. Four-week-old WT and *brak* (#5) mutants were inoculated with or without *B. cinerea*, and leaves were sampled at 1 dpi. A large number of genes were differentially expressed (fold change ≥2, $P < 0.05$) in both WT and *brak* mutants post inoculation (Fig. 8A; Dataset EV4). A total of 1656 transcripts were significantly increased in WT and 1413 in *brak* mutants. Among these, 584 transcripts were significantly upregulated in WT compared to *brak* mutants (Fig. 8B; Dataset EV5), defining 584 transcripts as *BRAK*-dependent *B. cinerea*-induced genes.

The top 20 enriched Gene Ontology (GO) categories in the *BRAK*-dependent *B. cinerea*-induced genes included "oxidation-reduction process" as the most significant, along with categories related to "protein phosphorylation" and "kinase activity" (Fig. 8C; Dataset EV6). These categories encompass genes encoding receptor-like kinases (RLKs), mitogen-activated protein kinase kinase kinases (MAPKKK), and calcium-dependent protein kinases (CPK). qRT-PCR analysis confirmed increased transcriptional abundance of representative genes in the "oxidation-reduction process" and "protein phosphorylation" categories in WT compared to *brak* mutants upon *B. cinerea* infection (Fig. 8D). These results indicate that phosphorylation is essential in BRAK-regulated defense.

## Discussion

In the natural environment, plants have to endure attacks from a range of pathogens. When defending against disease-causing agents, the redistribution of resources within the plant, particularly towards defense, often leads to the inhibition of growth. LRR-RLKs are a large gene family in plants that regulate various plant processes, including disease responses and growth. The function of only a small proportion of LRR-RLKs has been identified, with the majority remaining unexplored, particularly those implicated in promoting growth and defense against necrotrophic fungi. In this study, we identified a novel LRR-RLK BRAK from the tomato LRR-RLK family, which positively regulated seedling growth and resistance to *B. cinerea*. Molecular studies indicated that the reciprocal phosphorylation between BRAK and the PSK receptor PSKR1 was essential for growth and defense. Our findings suggest that the tomato LRR-RLK BRAK is a promising molecular target for genetic engineering programs aimed at generating *B. cinerea*-resistant vigorous crops. Several lines of evidence support this conclusion.

First, the response of the tomato LRR-RLK family to *B. cinerea* infection was investigated, and the differentially expressed genes in subfamily IV were all upregulated. Among the 13 members of subfamily IV, BRAK exhibited the highest upregulation (Fig. 1), which is further confirmed by our transcriptomic data (Dataset EV4; Appendix Fig. S4). By generating *brak* mutants and OE-*BRAK* plants, we first reported that BRAK significantly enhanced resistance to *B. cinerea* and promoted tomato growth and fruit production (Fig. 2). Previous studies have shown that its Arabidopsis homologue NILR1 (AT1G74360) is required for triggering immune signaling against nematodes (Mendy et al, 2017; Huang et al, 2023). Notably, some members of subfamily IV have been reported to function in either immune response to pathogens or plant

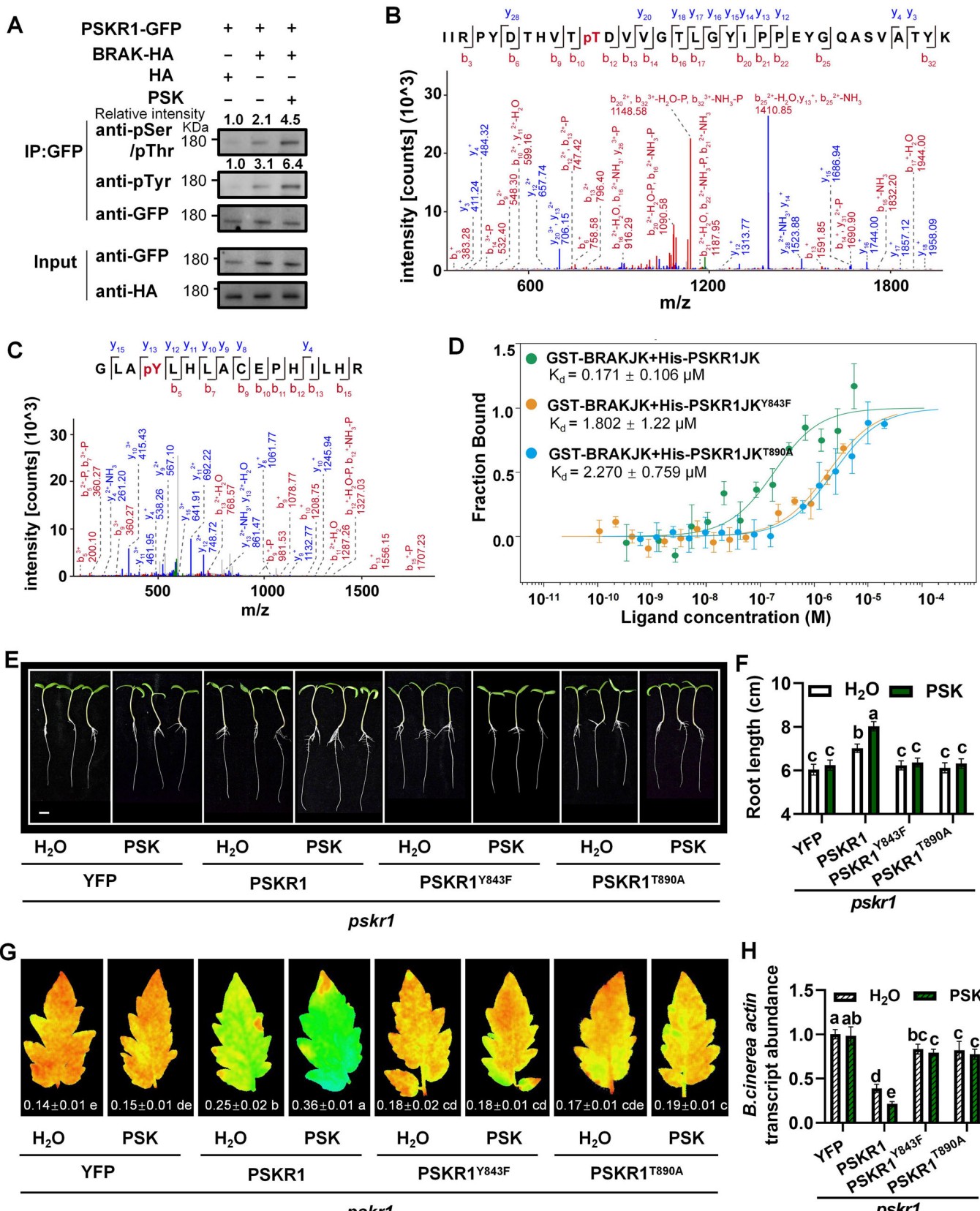

◄ **Figure 7. BRAK phosphorylates PSKR1 at Y843 and T890 to regulate tomato growth and defense against *B. cinerea*.**

(A) PSK induces the phosphorylation of PSKR1 by BRAK in vivo. PSKR1-GFP was co-expressed with BRAK-HA (empty vector as a negative control) in *N. benthamiana* leaf. And 10 μM PSK or H₂O was injected into the leaf before sampling. The phosphorylated PSKR1 protein was precipitated by GFP-Trap, and then immunoblotted with anti-pSer/Thr and anti-pTyr antibodies. (B, C) In vitro mass spectrometric analysis of PSKR1 phosphorylation site by BRAK. PSKR1 T890 site (B) and Y843 site (C). Threonine and tyrosine residues in the PSKR1 are highlighted in red. (D) In vitro interaction between BRAKJK and WT or phospho-mutated PSKR1JK revealed using MST. BRAKJK-GST was regarded as the target, and the WT or phospho-mutated PSKR1JK protein was used as the ligand. (E, F) Phenotype and index of plant seedling growth. The representative image (E) and root lengths data (F) of *pskr1* mutant transiently overexpressed with YFP (as a negative control), wild-type PSKR1 or phospho-dead PSKR1 grown on 1/2 MS medium with or without 1 μM PSK for 5 days. Bar = 1 cm. (G, H) Phenotype and index of plants infected by *B. cinerea*. Chlorophyll fluorescence of ΦPSII (G), and *B. cinerea* actin transcript abundance (H) in leaves at 1 dpi. Leaves of 4-week-old tomato were treated with 10 μM PSK or H₂O 12 h before *B. cinerea* inoculation. Data in (F–H) are presented as mean values ± SD, $n = 8$ different tomato seedings in (F), $n = 8$ leaves from different plants in (G), $n = 4$ independent pooled samples with each sample being from two plants in (H). Different letters depict statistically significant differences, as analyzed by one-way ANOVA with Tukey's HSD post hoc test ($P < 0.05$). Experiments in (A–D) were performed twice, and experiments in (E–H) were repeated three times with similar results. Source data are available online for this figure.

growth. However, unlike BRAK, these LRR-RLKs are incapable of simultaneously promoting growth and defense. For example, as a BR signaling receptor, BRI1 is required for growth and development but inhibits defense against various fungi in barley (Goddard et al, 2014). Another Arabidopsis LRR-RLK, BIR1 (BAK1-INTERACTING RECEPTOR-LIKE KINASE 1), negatively regulates plant resistance to bacterial pathogens (Liu et al, 2016), while BIR1 loss of function results in constitutive activation of cell death (Wang et al, 2011). BIR2 negatively regulates immunity to *Pst* DC3000 but does not affect BR-regulated growth (Halter et al, 2014). Likewise, SOBIR1 (Suppressor of BIR1-1) promotes defense against fungal pathogens such as *Phytophthora parasitica* and *Pyricularia oryzae* (Peng et al, 2015; Takahashi et al, 2018), whereas SOBIR1 overexpression leads to cell death (Gao et al, 2009). Thus, we identified a novel LRR-RLK, BRAK that positively contributes to both growth and defense against *B. cinerea*.

Second, BRAK-regulated growth and defense is associated with PSKR1. Through a Y2H screen, we identified a clone encoding the intracellular domain of an LRR-RLK family protein, PSKR1, that interacted with BRAK (Fig. 3). There are two PSKRs in plants, PSKR1 and PSKR2. It has been reported that PSKR1, rather than PSKR2, is the primary membrane-bound receptor of PSK, responsible for regulating plant growth and defense (Hartmann et al, 2013; Mosher et al, 2013; Zhang et al, 2018). PSKR1 promotes resistance to fungal pathogens such as *Alternaria brassicicola* and *Sclerotinia sclerotiorum* in Arabidopsis (Mosher et al, 2013) and Verticillium wilt in cotton (Zhang et al, 2022). Additionally, SOBIR1, a homolog of BRAK in subfamily IV, enhances defense against fungal pathogens such as *Pyricularia oryzae* in Arabidopsis (Takahashi et al, 2018). Therefore, it is possible that PSKR1 interacts with BRAK to jointly regulate resistance against a broader range of fungal pathogens beyond *B. cinerea*. A recent study revealed that the growth-promoting rhizosphere microbe *Pseudomonas fluorescens* induces PSKR1 expression in Arabidopsis roots, inhibiting salicylic acid immune signaling to promote the growth of *Pseudomonas* (Song et al, 2023). Moreover, PSKR1 is also involved in plant growth (Sauter, 2015), enhancing callus growth (Matsubayashi et al, 2006), adventitious root formation (Amano et al, 2007), root elongation (Kaufmann et al, 2021) in Arabidopsis. This is in line with our observation that PSKR1 promoted tomato seedling growth and fruit production (Figs. 4, 5 and EV1). Notably, PSKR1 interaction with BAK1 is required for PSK-mediated growth regulation (Ladwig et al, 2015; Wang et al, 2015). However, the potential involvement of BAK1 in the PSKR1/BRAK-regulated growth and resistance remains unclear, providing a ground for future exploration.

The very large number of LRR-RLKs form complex, contributing to the diversification and amplification of signaling pathways regulating growth, development, and immune responses (Smakowska-Luzan et al, 2018). For instance, the cell-surface receptor heteromer MDIS1-MIK functions in plant reproductive growth (Wang et al, 2016), while LRR-RLK IOS1 interacts with FLS2 and ERF to trigger immune responses (Yeh et al, 2016). Moreover, the *brak*, *pskr1* single and double mutants exhibited comparable growth and defense phenotypes, as well as responses to PSK treatment (Fig. 4). Additionally, PSK-induced genes showed almost no response in *brak* mutants upon PSK treatment. *BRAK* silencing in OE-PSKR1 plants impaired growth and immunity (Fig. 5). Thus, it raises the possibility that BRAK interacts with PSKR1, with BRAK likely functioning downstream of PSKR1 within the PSK signaling pathway to regulate plant seedling growth and defense against *B. cinerea*. These findings underscore the importance of the BRAK-PSKR1 complex in shaping plant growth and immune responses.

Third, reciprocal phosphorylation between BRAK and PSKR1 was critical for growth and defense. Our data show that BRAK and PSKR1 phosphorylated each other at threonine and tyrosine residues. Phosphorylation of BRAK at T1027 and Y1048, and PSKR1 at Y843 and T890, promotes their interaction, which is crucial for the fine regulation of plant growth and resistance to *B. cinerea* (Figs. 6, 7 and EV2). While plant RLKs are generally classified as serine/threonine kinases, we observed that BRAK and PSKR1 may also possess tyrosine kinase activity. Similarly, BAK1, classically defined as a serine/threonine kinase, is found to autophosphorylate and phosphorylate BIK1 on tyrosine residues (Lin et al, 2014). Phosphorylation of LRR-RLKs plays a crucial role in plant growth and immunity. In Arabidopsis, phosphorylation of the juxtamembrane and C-terminal domains of PSKR1 impacts plant growth, and regulates its receptor activity (Kaufmann et al, 2017). The BRI1^Y1052F site-directed mutant only slightly rescues the dwarf phenotype of the *bri1* mutant (Oh et al, 2009). Mutations in FLS2 S938 dissect signaling activation in FLS2-mediated immunity (Cao et al, 2013). Moreover, reciprocal phosphorylation is a common feature among RLKs. For example, upon BL stimulation, BRI1 associates with BAK1 and phosphorylates it on the kinase domain, subsequently leading to BAK1 phosphorylating BRI1, enhancing BR signaling (Wang et al, 2008). Therefore, we hypothesize that PSKR1 is first phosphorylated upon PSK perception and subsequently phosphorylates BRAK. The activated BRAK then enhances signaling output through reciprocal PSKR1 phosphorylation. Our findings reveal a new LRR-RLK complex that

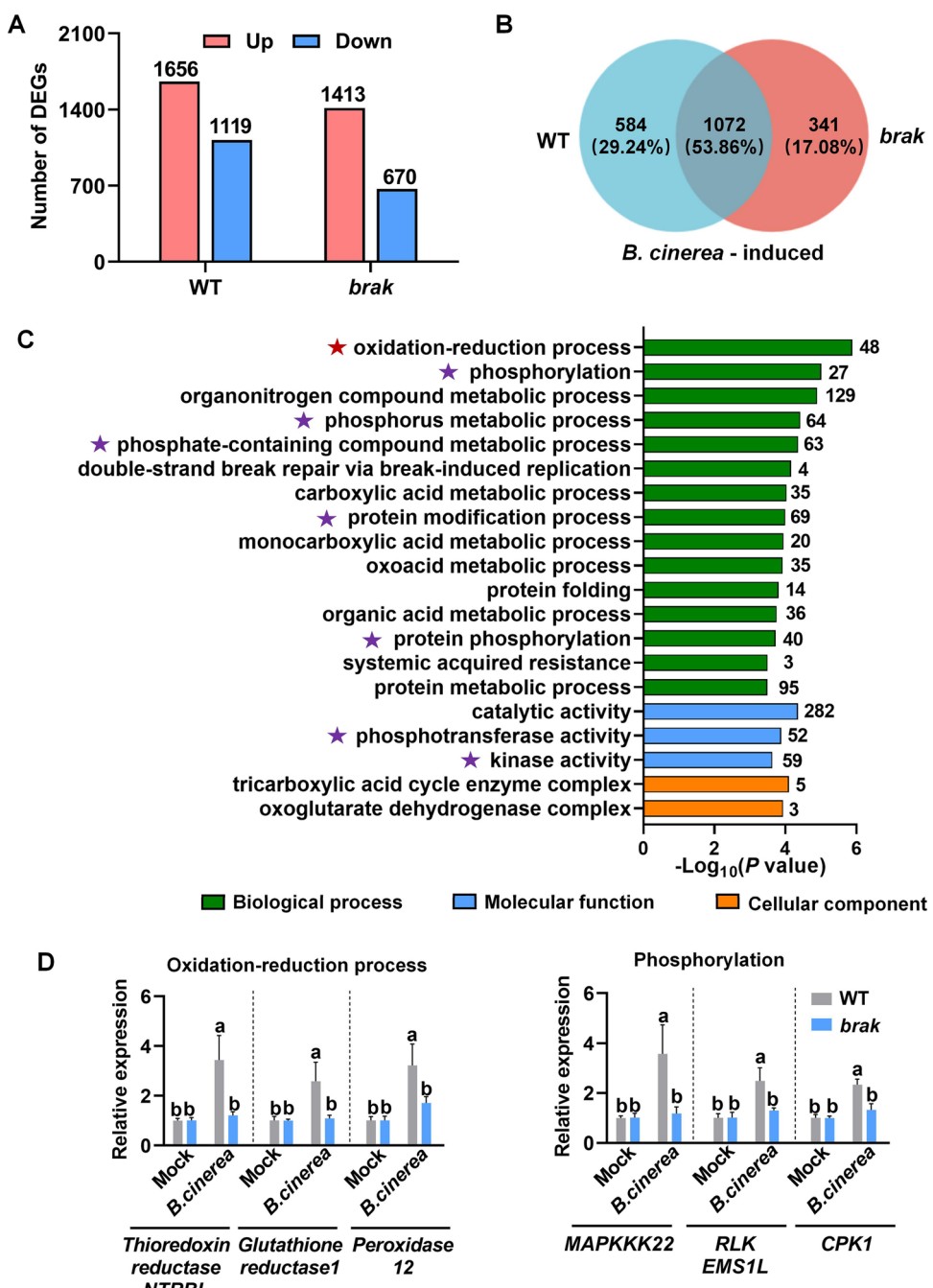

**Figure 8. BRAK-regulated defense against *B. cinerea* is associated with the oxidation-reduction process and phosphorylation.**

(A) Numbers of differentially *B. cinerea*-changed genes (*B. cinerea* vs Mock, fold change ≥2 and $P < 0.05$) in WT and *brak#5* mutants. Leaf samples were collected 1 day-post-inoculation (dpi) with *B. cinerea*. (B) Venn diagram showing the number of *B. cinerea*-induced genes (fold change ≥2 and $P < 0.05$) in WT and *brak* mutants. (C) Enriched GO categories of BRAK-dependent *B. cinerea*-induced genes. The numbers of enriched genes in each GO term were shown at the right of column. Red stars represent oxidation-reduction related GO term, while purple stars represent phosphorylation-related GO term. (D) qRT-PCR analysis confirming expression of selected genes from specific GO items. Leaves were sampled 24 h with or without *B. cinerea* inoculation. Data represent means ± SD, $n = 3$ independent pooled samples with each sample being from two plants. Different letters depict statistically significant differences, as analyzed by one-way ANOVA with Tukey's HSD post hoc test ($P < 0.05$). Source data are available online for this figure.

positively functions in plant growth and defense, providing novel insights into PSK signaling pathways.

This discovery raises intriguing questions regarding the trade-off between growth and defense. It is commonly believed that higher resistance compromises plant growth. However, we observed that BRAK and PSKR1 promote growth and defense simultaneously. One possible explanation is based on previous research (Hu et al, 2023) reporting that in the absence of pathogens,

PSKR1 is moderately ubiquitinated and degraded by the E3 ubiquitin ligase PUB12/13, preventing continuous activation of the immune response, thus maintaining a good growth state. Upon pathogens invasion, PUB12/13 dissociates from PSKR1, triggering downstream defense responses. This fine-tuned regulation may be key to boosting both resistance to pathogens and growth, even in crop production.

In conclusion, we identified a novel tomato LRR-RLK, BRAK. Further in-depth study revealed that BRAK interacted with and reciprocally phosphorylated the peptide receptor PSKR1, playing a positive role in tomato growth and defense against *B. cinerea*. This research extends our understanding of the function of LRR-RLKs, and opens new possibilities for producing *B. cinerea*-tolerant plants with vigorous growth and yield.

# Methods

### Reagents and tools table

| Reagent/resource | Reference or source | Identifier or catalog number |
| --- | --- | --- |
| **Experimental models** | | |
| Condine Red (*S. lycopersicum*) | Zhang et al (2018) | N/A |
| BO5-10 (*B. cinerea*) | Zhang et al (2018) | N/A |
| **Recombinant DNA** | | |
| pCAMBIA1301-BRAK | This study | N/A |
| pCAMBIA1301-PSKR1 | Ding et al (2023) | N/A |
| pFGC1008-BRAK | This study | N/A |
| pFGC1008-PSKR1 | Ding et al (2023) | N/A |
| pAC402-BRAK | This study | N/A |
| pAC402-PSKR1 | This study | N/A |
| pFGC1008-PSKR1JK | This study | N/A |
| pAC402-BRAKJK | This study | N/A |
| pGADT7-PSKR1JK | This study | N/A |
| pGBKT7-BRAKJK | This study | N/A |
| pTRV2-BRAK | This study | N/A |
| pDONR-BRAK | This study | N/A |
| pDONR-PSKR1 | Ding et al (2023) | N/A |
| pDONR-BRAK$^{T1027A}$ | This study | N/A |
| pDONR-BRAK$^{Y1048F}$ | This study | N/A |
| pDONR-BRAK$^{K830E}$ | This study | N/A |
| pDONR-PSKR1$^{Y843F}$ | This study | N/A |
| pDONR-PSKR1$^{KM}$ | This study | N/A |
| pDONR-PSKR1$^{T890A}$ | This study | N/A |
| pGWB417-BRAK$^{T1027A}$ | This study | N/A |
| pGWB417-BRAK$^{KM}$ | This study | N/A |
| pGWB417-PSKR1$^{Y843F}$ | This study | N/A |
| pGWB417-PSKR1$^{T890A}$ | This study | N/A |
| pGEX-4T-1-PSKR1JK | This study | N/A |
| pGEX-4T-1-BRAKJK | This study | N/A |

| Reagent/resource | Reference or source | Identifier or catalog number |
| --- | --- | --- |
| pET28a-BRAKJK | This study | N/A |
| pET28a-BRAKJK$^{T1027A}$ | This study | N/A |
| pET28a-BRAKJK$^{Y1048F}$ | This study | N/A |
| pET28a-PSKR1JK | Ding et al (2023) | N/A |
| pET28a-PSKR1JK$^{Y843F}$ | This study | N/A |
| pET28a-PSKR1JK$^{T890A}$ | This study | N/A |
| p2YN-BRAK | This study | N/A |
| p2YC-PSKR1 | Ding et al (2023) | N/A |
| p2YN-GGC1 | Wang et al (2022) | N/A |
| p2YC-GGC1 | Wang et al (2022) | N/A |
| **Antibodies** | | |
| Anti-HA | ThermoFisher Scientific | 26183 |
| Anti-MYC | Merck | clone 9E10 |
| Anti-GFP | ThermoFisher Scientific | GF28R |
| Anti-pSer/Thr | ECM Biosciences | PM3801 |
| Anti-pTyr | GenScript | A01819 |
| **Oligonucleotides and other sequence-based reagents** | | |
| PCR primers | This study | Table EV7 |
| **Chemicals, enzymes and other reagents** | | |
| PSK | Iris Biotech | N/A |
| MS Basal Medium with Vitamins | PhytoTech Labs | M519 |
| Inositol | Sigma | PHR1351 |
| MES | Sangon Biotech | A610611-0250 |
| Cellulose R10 | Yakult | N/A |
| Macerozyme R10 | Yakult | N/A |
| BSA | Fdbio science | FD0030 |
| Glucose | Sigma | Y0001745 |
| PEG 4000 | Sigma | 1546569 |
| ClonExpress II One-Step Cloning Kit | Vazyme Biotech | C112-02 |
| Protease inhibitor | Roche | 4693116001 |
| RNA extraction kits | Eazy-do Biotech | DR0406050 |
| ReverTra Ace quantitative (qPCR) RT Kits | Toyobo | FSQ-201 |
| SYBR Green PCR Master Mix | Takara | 4913850001 |
| Mut Express II Fast Mutagenesis Kit V2 | Vazyme Biotech | C214-02 |
| Mannitol | Sigma | M0200000 |
| Protein Labeling Kit RED-NHS 2nd Generation | NanoTemper Technologies | MO-L011 |
| GFP-Trap Magnetic Agarose | Chromotek | gtma |
| **Software** | | |
| GraphPad Prism 8.0 | https://www.graphpad.com | N/A |
| Affinity Analysis 3 | NanoTemper | N/A |

| Reagent/resource | Reference or source | Identifier or catalog number |
|---|---|---|
| **Other** | | |
| Illumina NovaSeq X | Illumina | N/A |
| Illumina HiSeq2000 | Illumina | N/A |

## Plant material and growth conditions

All tomato (*Solanum lycopersicum* L.) lines used were in the Condine Red wild-type (WT) background. Seeds were planted to a combination of vermiculite and peat (1: 3, v/v) in a greenhouse at 22/20 °C (day/night) with a 14/10 h (day/night) photoperiod. The lighting was 400 µmol m$^{-2}$ s$^{-1}$, and the relative humidity was 80%.

Stable knock-out and overexpression lines were generated by gene editing and overexpression approaches, respectively. CRISPR/Cas9 gene editing was carried out using the previous methods (Hu et al, 2021). To construct *BRAK*-overexpressing transgenic tomato plants, the full-length CDS of *BRAK* was cloned from the tomato genome and then inserted into the pFGC1008 vector with a hemagglutinin (HA) tag under the 35S CaMV promoter. The plasmid was transformed into tomato by *Agrobacterium tumefaciens* GV3101 strain. The *pskr1 brak* double mutants were produced by crossing and selecting homozygous progeny by PCR analysis. The point-mutant plasmids were generated by Mut Express II Fast Mutagenesis Kit V2 (Vazyme Biotech). The transient overexpression assays in tomato plants were performed by vacuum-infiltrated with *A. tumefaciens* C58C1 strain. The primers for vector construction and mutant plant verification were listed in Dataset EV7. The transgenic and transient overexpression plants were identified via western blot using an anti-HA antibody (ThermoFisher Scientific, 26183) and an anti-MYC antibody (Merck, clone 9E10), respectively.

## Pathogen treatment and disease symptom assays

The *B. cinerea* BO5-10 strain was used in all pathogen inoculation experiments. The abaxial and adaxial leaves of the 4-week-old tomato were sprayed with *B. cinerea* suspension at a density of $2 \times 10^5$ spores mL$^{-1}$, and media buffer was used as mock inoculations. For the PSK pretreatment assay, leaves were sprayed with 10 µM PSK or H$_2$O 12 h before inoculation (Zhang et al, 2018).

Disease symptoms were determined by *B. cinerea actin* mRNA accumulation quantification through qRT-PCR, and chlorophyll fluorescence measurement with the imaging-PAM chlorophyll fluorometer (IMAG-MAXI, Heinz Walz). The quantum efficiency of light-adapted leaves (ΦPSII) was calculated as the formula $Fm'-F/Fm'$. *F* means actual fluorescence intensity at any time, and $Fm'$ means light-adapted maximum fluorescence (Genty et al, 1989). ΦPSII, as the actual photochemical efficiency, reflects the conversion of light energy in leaves. When plants are affected by diseases, photosynthesis is inhibited, leading to a decrease in ΦPSII (Roháček et al, 2008).

## Growth phenotype determination

Plant height was measured using a vernier caliper from the cotyledon node to the apical growth point in four-week-old tomato seedlings. For the plate experiments, tomato seeds were sterilized for 30 s in 75% (v/v)

ethanol, 15 min in 10% (v/v) sodium hypochlorite, and washed in autoclaved water before being cultured at 28 °C in a shaker run at 200 rpm for 2 days. When sterilized seeds germinated, seedlings with uniform growth were selected and placed on vertically oriented square plates containing 1/2 MS medium (4.3 g L$^{-1}$ MS Basal Medium with Vitamins, 0.01% inositol, 1% sucrose, 0.8% agarose, pH 5.7 with KOH) with or without 1 µM PSK.

## Protoplast isolation and transfection

Protoplasts were isolated from the well-grown tomato cotyledons (about 2–3 weeks old). Approximately 15 cotyledons were placed in a 6-cm Petri dish with 10 mL digestion solution (20 mM MES, 1% cellulose R10 and 0.25% macerozyme R10, 20 mM KCl, and 0.4 M mannitol, 10 mM CaCl$_2$, 0.1% BSA, pH 5.7) without lower epidermis. The material was incubated at room temperature for 3 h. The digested solution was filtered through a 70-µm nylon mesh. The sample was centrifuged at $100 \times g$ for 2 min to collect the protoplasts, which were then resuspended with 2 mL W5 solution (2 mM MES, 5 mM KCl, 154 mM NaCl, 125 mM CaCl$_2$, and 10 mM glucose, pH 5.7). The protoplasts were purified in 0.55 M sucrose solution and then washed with W5 solution. The protoplasts were transferred to MMG buffer (0.4 M mannitol, 4 mM MES, 15 mM MgCl$_2$) and adjusted to a concentration of $1 \times 10^5$ cells/mL.

The protoplasts were transfected with plasmids by PEG-calcium transfection buffer (40% PEG 4000, 0.2 M Mannitol, 100 mM CaCl$_2$). A 200 µL protoplast was combined with 20 µL of vector and mixed carefully. The same volume (220 µL) of PEG-calcium transfection buffer was added to the sample, mixed, and incubated for 15 min. To end the reaction, 880 µL of W5 was added, and the sample was mixed well. Transfected protoplasts were collected by centrifugation at $100 \times g$ for 3 min. The protoplasts were transferred to 1 mL of W5 in a 12-well plate and incubated in the dark at room temperature for 16 h to observe fluorescence.

## Y2H screening and assay

The coding sequences of BRAKJK, the juxtamembrane and kinase domain of BRAK, were cloned into the pGBKT7 vector as bait and transformed into yeast AH109 strain. Following mating, colonies were grown on SD medium lacking His, Trp, Ade, and Leu (SD-His-Trp-Ade-Leu).

For the Y2H assay, pGBKT7-BRAKJK and pGADT7-PSKR1JK were transformed into AH109, and then grown on SD-Trp-Leu medium, and SD-His-Trp-Ade-Leu medium with varying dilution amounts to further confirm probable positive clones.

## BiFC and Co-IP assays

To generate the vector for BiFC assays, the coding sequences were cloned into p2YC and p2YN using Clonexpress II one-step cloning kit (Vazyme Biotech). The primers used are listed in Dataset EV7. *A. tumefaciens* GV3101 strains carrying the pairs of binary constructs were transiently expressed in *N. benthamiana* leaves for BiFC assays. The determination of YFP was performed with a Zeiss LSM 780 confocal microscope 48 h after infiltration. The excitation/emission wavelengths were set up as 514 nm/520–560 nm.

For co-IP assays, the coding sequence of PSKR1 and PSKR1JK were cloned into pFGC1008-HA, and the sequence of BRAK and

BRAKJK were cloned into pAC402-GFP. Pairs of binary vectors containing either an HA or GFP tag were co-expressed in *N. benthamiana* leaves. Two days later, the leaves were injected with 10 μM PSK or ddH$_2$O 2 h before sampling. Leaf samples were taken and ground to powder in liquid nitrogen. And then 1 mL IP buffer (50 mM Tris-HCl pH 7.5, 150 mM NaCl, 5 mM EDTA, 0.5% Triton X-100; 1× protease inhibitor (Roche), 0.1 mM DTT, 2 mM Na$_3$VO$_3$ and 2 mM NaF added before using) was added into the sample. After centrifugation at 4 °C, GFP-Trap Magnetic Agarose (Chromotek) was added to the supernatant. The beads were washed 5 times with washing buffer (5 mM EDTA, 50 mM Tris-HCl pH 7.5, 150 mM NaCl, 0.1% Triton X-100) after rotation for 3 h at 4 °C. The beads were collected for immunoblot with an anti-HA (ThermoFisher Scientific, 26183) or anti-GFP (ThermoFisher Scientific, GF28R) antibody to detect the immunoprecipitated proteins.

## Microscale thermophoresis (MST) analysis

To confirm the protein interaction, fused proteins were prokaryotically expressed. GST-BRAKJK and GST-PSKR1JK were labeled with RED-NHS (NanoTemper Technologies, MO-L011). The protein mixtures were incubated with an interaction buffer (1× PBS-T, 2 mM DTT and 0. 05% Tween-20). The samples were then loaded into Monolith Capillaries (NanoTemper Technologies, MO-K022) using 40% LED excitation power and medium MST power at 25 °C. The Kd values from triplicate experiments were calculated using MO. Affinity Analysis 3 software provided by NanoTemper.

## LC-MS/MS analysis and phosphorylation assays

The in vitro phosphor-sites identification was performed with LC-MS/MS analysis as reported previously (Ding et al, 2023). The purified substrate protein and the kinase protein were incubated in kinase buffer (25 mM Tris-HCl, pH 7.5, 10 mM MgCl$_2$, 0.1 mM CaCl$_2$, 1 mM DTT, and 0.2 mM ATP) for 3 h at room temperature with gentle shaking, and the reaction was stopped by adding 4× SDS loading buffer. Then, the substrate proteins were separated and collected by SDS-PAGE for further digestion with trypsin overnight. The phosphopeptides were analyzed with LTQ Orbitrap Elite (ThermoFisher).

For in vivo phosphorylation assay, the corresponding binary vectors with either a GFP or HA tag were co-expressed in *N. benthamiana* leaves. Two days later, the proteins were extracted and incubated in IP buffer with GFP-Trap Magnetic Agarose (Chromotek).

The phosphorylation of target proteins from in vitro and in vivo was analyzed by immunoblotting with an anti-pSer/Thr antibody (ECM Biosciences, PM3801) or anti-pTyr antibody (GenScript, A01819).

## Generation of virus-induced gene-silencing seedlings

VIGS was performed on germinated tomato seedlings by vacuum-infiltrated with *Agrobacterium*. For the plate experiment, tomato seeds were first surface-sterilized in 75% (v/v) ethanol for 30 s, and then transferred to 10% (v/v) sodium hypochlorite for 15 min, and washed five times with autoclaved H$_2$O under sterile conditions. Two days later, the germinated seeds were vacuum-infiltrated with a mix of pTRV1 and pTRV2 vectors and then washed with autoclaved H$_2$O ten times, and finally laid out on 13 cm × 13 cm plastic plates containing 1/2 MS medium under sterile conditions. Based on the online VIGS tool (available at http://vigs.solgenomics.net/), partial cDNA fragments of tomato *BRAK* were cloned into pTRV2. The empty pTRV2 was used as a negative control. Silencing efficiency was determined using qRT-PCR with primers shown in Dataset EV7. Plants with significant silencing efficiency were chosen for further study.

## Transcript and RNA-seq analysis

Total RNA was extracted with RNA extraction kits (Eazy-do Biotech, China), and reverse transcription was performed by ReverTra Ace quantitative (qPCR) RT Kits (Toyobo, Japan). Quantitative real-time PCR (qRT-PCR) was analyzed using SYBR Green PCR Master Mix (Takara) with a Light Cycler 480 real-time PCR system (Roche, Switzerland). The reaction buffer and PCR conditions were the same as previously described (Ding et al, 2021). The primers used are listed in Dataset EV7.

Six 4-week-old WT and *brak*#5 plants were inoculated with *B. cinerea*, and 3 independent pooled samples from two plant leaves at 1 dpi were sequenced. Twenty-four 3-day-old WT and *brak*#5 seedlings were treated with 1 μM PSK on 1/2 MS media, with H$_2$O as a control, and three independent pooled samples from eight plant roots 12 h post-treatment were sequenced. Library construction, RNA sequencing, and bioinformatics analyses for leaf samples were performed by Majorbio (Shanghai, China) using the Illumina HiSeq2000 platform, while root samples were processed by Baimaike (Shandong, China) using the Illumina NovaSeq X Plus platform. Clean reads were mapped to the *Solanum lycopersicum* genome database (https://www.solgenomics.net/, version SL4.0).

## Statistical analysis

For each determination, at least three independent biological replicates were sampled. The experiments were independently performed two or three times. All of the bar chart plots were processed and presented using GraphPad Prism 8. The obtained data were subjected to analysis of variance using SAS 8.0 (SAS Institute, Cary, NC, USA), and means were compared using Tukey's test at the 5% level.

# Data availability

RNA-seq data can be accessed in the NCBI database, with leaf samples under BioProject accession number PRJNA1041583 (https://www.ncbi.nlm.nih.gov/bioproject/PRJNA1041583) and root samples under PRJNA1144371 (https://www.ncbi.nlm.nih.gov/bioproject/PRJNA1144371).

The source data of this paper are collected in the following database record: biostudies:S-SCDT-10_1038-S44318-024-00278-z.

# Peer review information

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

## Acknowledgements

We thank Prof. Changtian Pan (Zhejiang University) for assistance with tomato protoplast isolation and transfection. We are grateful for the help of Dr. Yang Li (University of Michigan) who provided technical support with phylogenetic analysis. This work was supported by the National Key Research and Development Program of China (2023YFD2300701), the National Natural Science Foundation of China (32172650, 32302639, 32430092), the Starry Night Science Fund of Zhejiang University Shanghai Institute for Advanced Study (SN-ZJU-SIAS-0011) and China Postdoctoral Science Foundation (2022M722800, 2023T160572).

## Author contributions

**Shuting Ding**: Conceptualization; Funding acquisition; Investigation; Visualization; Methodology; Writing—original draft; Writing—review and editing. **Shuxian Feng**: Investigation; Visualization; Writing—original draft. **Shibo Zhou**: Investigation; Visualization. **Zhengran Zhao**: Investigation. **Xiao Liang**: Investigation. **Jiao Wang**: Investigation. **Ruishuang Fu**: Investigation. **Rui Deng**: Writing—review and editing. **Tao Zhang**: Technical support. **Shujun Shao**: Technical support. **Jingquan Yu**: Resources. **Christine H Foyer**: Resources. **Kai Shi**: Conceptualization; Resources; Supervision; Funding acquisition; Writing—original draft; Project administration; Writing—review and editing.

Source data underlying figure panels in this paper may have individual authorship assigned. Where available, figure panel/source data authorship is listed in the following database record: biostudies:S-SCDT-10_1038-S44318-024-00278-z.

## Disclosure and competing interests statement

The authors declare no competing interests.

# Expanded View Figures

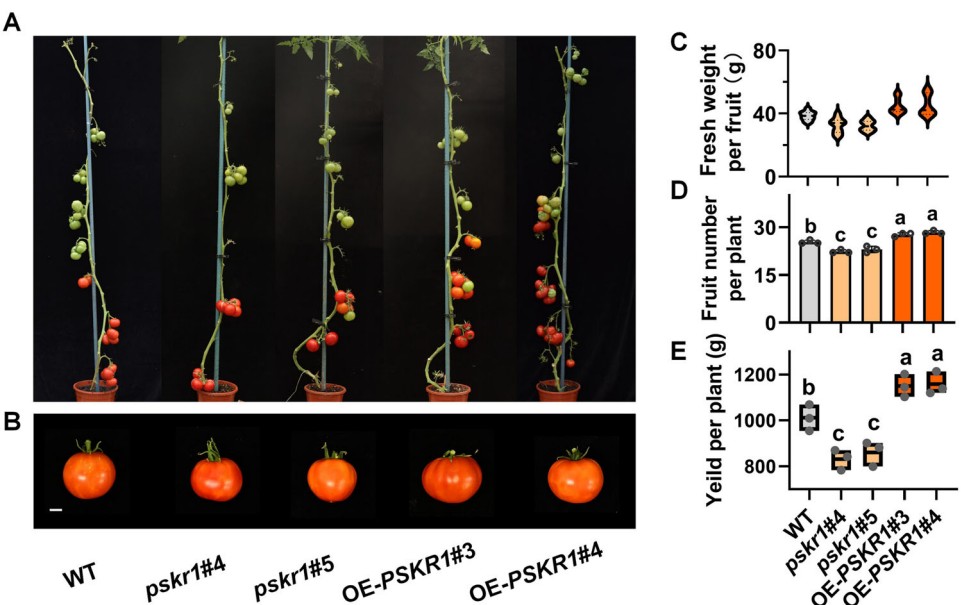

**Figure EV1. The effect of PSKR1 on tomato fruit phenotype.**

(A, B) Representative plant fruit phenotype in the reproductive stage. Bar, 1 cm (B). (C–E) Fresh weight per fruit (C), Fruit number per plant (D) and yield (E). Data in (C–E) are presented as mean values ± SD, $n = 10$ different tomato fruits in (C), $n = 3$ individual plants in (D), $n = 3$ individual plants in (E). For boxplots, the center line in the box indicates median, dots represent data, limits represent upper and lower quartiles. Different letters depict statistically significant differences, as analyzed by one-way ANOVA with Tukey's HSD post hoc test ($P < 0.05$). Plants were grown in the greenhouse. Source data are available online for this figure.

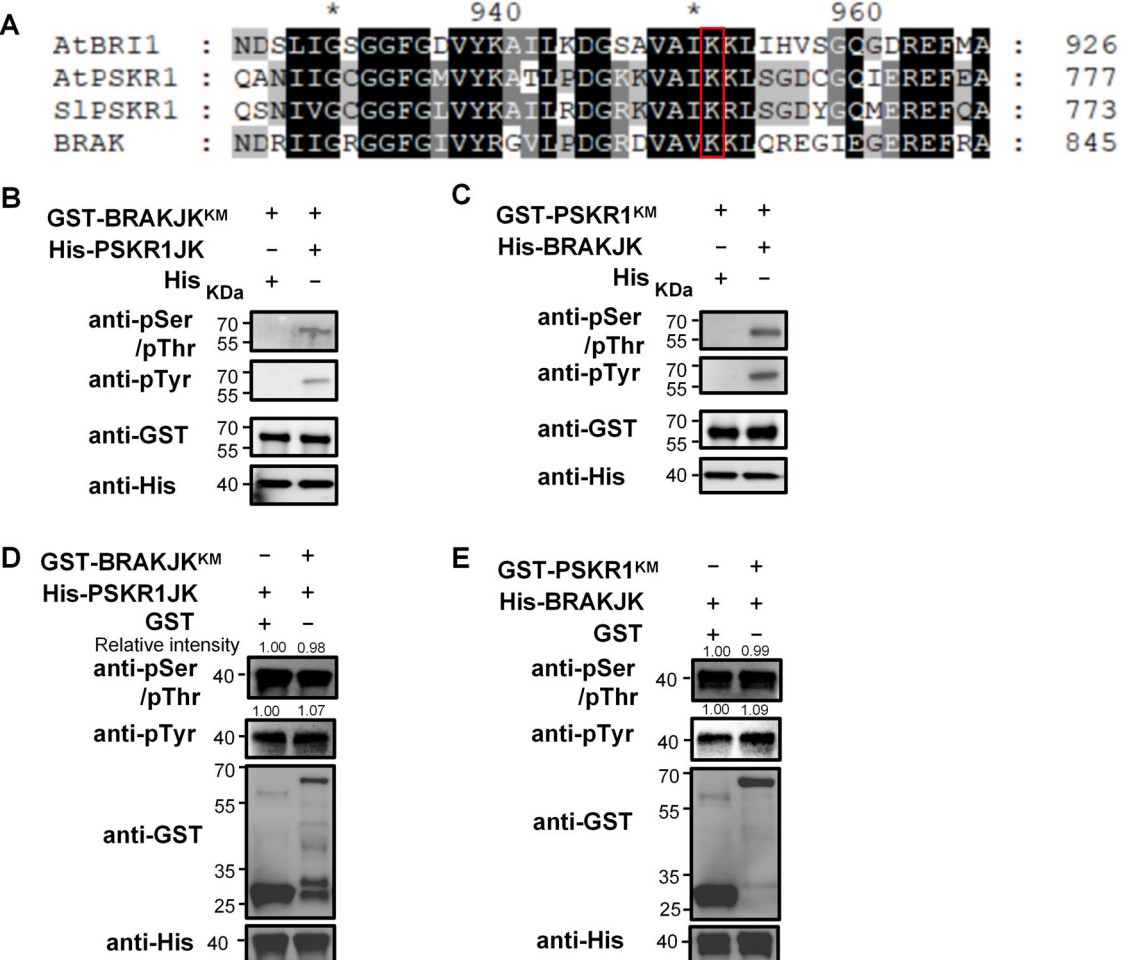

**Figure EV2. BRAK and PSKR1 phosphorylate each other in vitro.**

(A) Amino acid alignment among Arabidopsis BRI1, PSKR1, tomato PSKR1 and BRAK. The alignment was generated using Clustal X. Identical or similar amino acids are highlighted by black and gray backgrounds, respectively. The conserved lysine (K) in the ATP-binding site is boxed in red. (B) PSKR1 phosphorylate BRAK in vitro. The kinase assay was performed by incubating His or His-PSKR1JK as the kinase and GST-BRAKJK$^{KM}$. Phosphorylation is shown by western blot with anti-pSer/Thr and anti-pTyr antibodies. The protein loading control is shown by western blot with anti-GST and anti-His antibodies. (C) BRAK phosphorylate PSKR1 in vitro. The kinase assay was performed by incubating His or His-BRAKJK as the kinase and GST-PSKR1JK$^{KM}$. Phosphorylation is shown by western blot with anti-pSer/Thr and anti-pTyr antibodies. The protein loading control is shown by western blot with anti-GST and anti-His antibodies. (D) The effect of BRAK on the autophosphorylation of PSKR1 in vitro. The kinase assay was performed by incubating His-PSKR1JK and GST-BRAKJK$^{KM}$. Phosphorylation is shown by western blot with anti-pSer/Thr and anti-pTyr antibodies. The protein loading control is shown by western blot with anti-GST and anti-His antibodies. (E) The effect of PSKR1 on the autophosphorylation of BRAK in vitro. The kinase assay was performed by incubating His or His-BRAKJK as the kinase and GST-PSKR1JK$^{KM}$. Phosphorylation is shown by western blot with anti-pSer/Thr and anti-pTyr antibodies. The protein loading control is shown by western blot with anti-GST and anti-His antibodies. Source data are available online for this figure.

