## [Peer Review File · The EMBO Journal]

A novel LRR receptor-like kinase BRAK reciprocally phosphorylates PSKR1 to enhance growth and defense in tomato

Shuting Ding, Shuxian Feng, Shibo Zhou, Zhengran Zhao, Xiao Liang, Jiao Wang, Ruishuang Fu, Rui Deng, Tao Zhang, Shujun Shao, Jingquan Yu, Christine H. Foyer, and Kai Shi

Corresponding author(s): Kai Shi (kaishi@zju.edu.cn)

Review Timeline:

Submission Date:	23rd Feb 24
Editorial Decision:	11th Apr 24
Revision Received:	9th Aug 24
Editorial Decision:	13th Sep 24
Revision Received:	2nd Oct 24
Accepted:	4th Oct 24

Editor: William Teale

Transaction Report:

Dear Dr. Shi,

Thank you for submitting your manuscript for consideration by the EMBO Journal. It has now been seen by two referees whose comments are shown below.

Given the referees' positive recommendations, I would like to discuss with you the feasibility of submitting a revised version of the manuscript that addresses the comments of both reviewers in full. I would be very interested in a much more careful dissection of the signalling pathways involved and a detailed assessment of yield gains. After you and your co-authors have had a chance to digest the reports, please let me know when a Zoom discussion would be convenient. I should add that it is EMBO Journal policy to allow only a single round of revision, and acceptance of your manuscript will therefore depend on the completeness of your responses in this revised version.

Thank you for the opportunity to consider your work for publication. I look forward to your revision.

Best regards,

William

William Teale, Ph.D.
Editor
The EMBO Journal

When submitting your revised manuscript, please carefully review the instructions below and include the following items:

- 1) a .docx formatted version of the manuscript text (including legends for main figures, EV figures and tables). Please make sure that the changes are highlighted to be clearly visible.
- 2) individual production quality figure files as .eps, .tif, .jpg (one file per figure).
- 3) a .docx formatted letter INCLUDING the reviewers' reports and your detailed point-by-point response to their comments. As part of the EMBO Press transparent editorial process, the point-by-point response is part of the Review Process File (RPF), which will be published alongside your paper.
- 4) a complete author checklist, which you can download from our author guidelines ([https://wol-prod-cdn.literatumonline.com/pb-assets/embo-site/Author Checklist%20-%20EMBO%20J-1561436015657.xlsx](https://wol-prod-cdn.literatumonline.com/pb-assets/embo-site/Author%20Checklist%20-%20EMBO%20J-1561436015657.xlsx)). Please insert information in the checklist that is also reflected in the manuscript. The completed author checklist will also be part of the RPF.
- 5) Please note that all corresponding authors are required to supply an ORCID ID for their name upon submission of a revised manuscript.
- 6) We require a 'Data Availability' section after the Materials and Methods. Before submitting your revision, primary datasets produced in this study need to be deposited in an appropriate public database, and the accession numbers and database listed under 'Data Availability'. Please remember to provide a reviewer password if the datasets are not yet public (see <https://www.embopress.org/page/journal/14602075/authorguide#datadeposition>). If no data deposition in external databases is needed for this paper, please then state in this section: This study includes no data deposited in external repositories. Note that the Data Availability Section is restricted to new primary data that are part of this study.

Note - All links should resolve to a page where the data can be accessed.

- 7) When assembling figures, please refer to our figure preparation guideline in order to ensure proper formatting and readability

in print as well as on screen:
<http://bit.ly/EMBOPressFigurePreparationGuideline>

8) For data quantification: please specify the name of the statistical test used to generate error bars and P values, the number (n) of independent experiments (specify technical or biological replicates) underlying each data point and the test used to calculate p-values in each figure legend. The figure legends should contain a basic description of n, P and the test applied. Graphs must include a description of the bars and the error bars (s.d., s.e.m.).

9) We would also encourage you to include the source data for figure panels that show essential data. Numerical data can be provided as individual .xls or .csv files (including a tab describing the data). For 'blots' or microscopy, uncropped images should be submitted (using a zip archive or a single pdf per main figure if multiple images need to be supplied for one panel). Additional information on source data and instruction on how to label the files are available at .

10) We replaced Supplementary Information with Expanded View (EV) Figures and Tables that are collapsible/expandable online (see examples in <https://www.embopress.org/doi/10.15252/emboj.201695874>). A maximum of 5 EV Figures can be typeset. EV Figures should be cited as 'Figure EV1, Figure EV2' etc. in the text and their respective legends should be included in the main text after the legends of regular figures.

12) Our journal encourages inclusion of *data citations in the reference list* to directly cite datasets that were re-used and obtained from public databases. Data citations in the article text are distinct from normal bibliographical citations and should directly link to the database records from which the data can be accessed. In the main text, data citations are formatted as follows: "Data ref: Smith et al, 2001" or "Data ref: NCBI Sequence Read Archive PRJNA342805, 2017". In the Reference list, data citations must be labeled with "[DATASET]". A data reference must provide the database name, accession number/identifiers and a resolvable link to the landing page from which the data can be accessed at the end of the reference. Further instructions are available at .

- a point-by-point response to the referees' comments, with a detailed description of the changes made (as a word file).
- a word file of the manuscript text.

- individual production quality figure files (one file per figure)
- a complete author checklist, which you can download from our author guidelines (<https://www.embopress.org/page/journal/14602075/authorguide>).
- Expanded View files (replacing Supplementary Information)
Please see out instructions to authors
<https://www.embopress.org/page/journal/14602075/authorguide#expandedview>

We realize that it is difficult to revise to a specific deadline. In the interest of protecting the conceptual advance provided by the work, we recommend a revision within 3 months (10th Jul 2024). Please discuss the revision progress ahead of this time with the editor if you require more time to complete the revisions. Use the link below to submit your revision:

Referee #1:

The authors discovered BRAK in tomato LRR-RLK subfamily IV based on gene expression patterns during *B. cinerea* infection, and immunoprecipitation experiments showed that BRAK is involved in defense responses as a PSKR co-receptor. This finding will add to our knowledge of plant immunity to *B. cinerea* in tomato and will help us to further understand the PSK signaling pathway. The paper is logically organized and conclusions are supported by rigid experimental data.

The authors have shown that BRAK mutant plants are insensitive to PSK in root elongation tests and resistance to *B. cinerea*. However, it is not clear from the data currently available whether BRAK is involved in all or only part of PSK signaling. I ask the authors to clarify this point by comparing the transcriptome of the WT treated with PSK with the transcriptome of the BRAK mutant treated with PSK.

PSKR1 is reported to be expressed throughout the plant body, but what is the expression pattern of BRAK, e.g. Promoter-GUS?

Although BRAK is a gene discovered in tomato, the Arabidopsis homologue AT1G74360 should be mentioned in the discussion. AT1G74360 has been shown in several papers to be involved in immune response.

Referee #2:

The authors have identified a class-IV LRR-RLK, BRAK, which is induced upon infection of tomato with *B. cinerea*. This protein appears directly involved in the defense response by making plants more resistant. Interestingly, BRAK also positively influences plant growth. BRAK was also found by Y2H to interact with PSKR1, an RLK known from Arabidopsis to bind PSK, a signal triggered during defense responses. Interestingly, als PSKR1 has a positive effect on defense responses and plant growth. The two RLKs seem to mutually activate each other by phosphorylation at Tyr and Thr residues and these phospho-sites are essential for the positive effect of the RLKs on defense and growth control.

The manuscript is very interesting in that it describes a positive effect of RLKs on defense and growth - which is rather uncommon. It also describes the interaction of RLKs and their effect on phosphorylation. Since there are a number of RLK-RLK interactions reported and described in detail (e.g. FER, BRI1, FLS2, BAK1....), the finding of an additional RLK-RLK interaction is not a major break-through. However, if these RLKs indeed boost both resistance to pathogens and growth, this would be a very interesting entry point for improvement of crop production.

Hence, to make this manuscript indeed a major step forward, the authors should show that their tomato plants indeed perform better / are more resistant and give at least a similar yield as the control plants. While the authors describe positive effects on seedling growth, they fail to quantify this supposed growth benefit at adult stages or in terms of fruit production.

Other comments:

Fig. 1B: the expression is relative to what?

Fig. 1C: the should provide a co-localization with a known plasma membrane-localized protein.

lane 150: where is the connection between quantum yield and defense/growth ? Please explain to the non-specialized reader.

Lane 161 ff: of all the clones that were identified, why did they specifically choose PSKR1? Could all the others not be

confirmed?

Lane 189: why was gene silencing used here? Is it that the crispr mutant in brak would not show the same phenotype? Or was it practical reasons?

Lane 193 and elsewhere: while the authors provide convincing evidence that PSKR1 and BRAK interact and are both important for PSK-induced responses, they still repeatedly state that BRAK functions downstream of PSK. What lets them come to this conclusion? Please explain.

Lane 212 and further down: since the phosphorylation of BRAK is important for its function (and the same for PSKR1), is the autophosphorylation of these proteins influenced/triggered by the interacting RLK? In other words, is autophosphorylation of PSRK1 equally effektiv when incubating with a kinase-dead BRAK? And vice versa, of course. That would provide an additional insight into the dynamics of this protein-protein interaction that should be technically feasible.

Fig. 8: in their RNA seq experiment, could the authors confirm the findings of the study by Courbier et al (which their RLK-identification was based on)?

Lane 337: the phosphorylation of BRAK and PSRK1 are clearly not essential for plant growth. In fact, both genes are dispensable since the double mutant seems to grow fine. It's the fine-regulation that is changed.

Minor comments:

The abstract should mention the plant species that this work is about. Also spell out MST.

In general, there is an inconsistent use of past and present tense. Description of experiments should be in past tense, statements on facts/established knowledge or interpretations, however, should be in present tense (since they hold true not only in the past but also now). This tremendously facilitates the reading.

Lane 47: progresses should probably be processes.

Lane 56: kinase activity is not necessary for all signaling activities of RLK.

Lane 73 and elsewhere: the plant species is not always clear. Here, it is BRI1 of barley? Or is the growth rate in Arabidopsis barely reduced?

Lane 74: What are CLVs? Explain.

Lane 93-96: this is not a correct sentence (grammar-wise).

Fig. 2G, 6D, and 7E: label the panels to make the interpretation easier. It is not clear whether the labelling of the panel below is also ment for the one above.

Fig. 5E and elsewhere: it is surprising to see such tiny SD with only 3 samples tested. Are the error bars perhaps SEM?

Dear Dr. William Teale and Referees,

We are pleased to submit our revised manuscript entitled “A novel LRR-RLK BRAK reciprocally phosphorylates PSKR1 to enhance growth and defense in tomato” (EMBOJ-2024-117048R) by Ding et al., for publication in the *EMBO Journal*. We appreciate the constructive comments and suggestions made by the Editors and the Reviewers that have helped us tremendously in improving our manuscript.

We have now revised the manuscript to address all the concerns raised by the Reviewers. The changes incorporated into the original manuscript have been highlighted with red color in the revised manuscript. The text of the manuscript has also been checked carefully, and we have been careful to adhere to the format of the *EMBO Journal*.

Our point-by-point responses to the Reviewers comments are detailed below. Notably, we found that both BRAK and PSKR1 can significantly enhance fruit yield, and the potential mechanisms for the simultaneous improvement of growth and defense was discussed in the discussion section. We believe that our revisions address all the points raised and that the revised version is acceptable for publication in the *EMBO Journal*. Please do not hesitate to contact us if further changes to the manuscript are required.

Sincerely,

Kai Shi, Ph. D

Professor, Department of Horticulture
Zhejiang University, China.
E-mail: kaishi@zju.edu.cn

August 9, 2024

Response to Referee #1:

The authors discovered BRAK in tomato LRR-RLK subfamily IV based on gene expression patterns during *B. cinerea* infection, and immunoprecipitation experiments showed that BRAK is involved in defense responses as a PSKR co-receptor. This finding will add to our knowledge of plant immunity to *B. cinerea* in tomato and will help us to further understand the PSK signaling pathway. The paper is logically organized and conclusions are supported by rigid experimental data.

Q1: The authors have shown that BRAK mutant plants are insensitive to PSK in root elongation tests and resistance to *B. cinerea*. However, it is not clear from the data currently available whether BRAK is involved in all or only part of PSK signaling. I ask the authors to clarify this point by comparing the transcriptome of the WT treated with PSK with the transcriptome of the BRAK mutant treated with PSK.

Thank you for your insightful comments. As suggested, we compared the transcriptomes of WT and *brak* mutant seedlings treated with 1 μ M PSK. Briefly, we found that 752 genes were induced by PSK treatment in WT, while only 28 gene expression were significantly changed in *brak* mutant. Among these, 749 transcripts were significantly altered in WT compared to *brak* mutants, indicating that the expression of 96.78% of PSK-regulated genes was BRAK-dependent. These findings shows that the lack of BRAK almost entirely prevents PSK from activating downstream responses, suggesting that BRAK is involved in almost all of PSK signaling. The results are now reported in the revised version of our manuscript in **Line 205-213**. The data are shown in **Figure 5G-H**.

Q2: PSKR1 is reported to be expressed throughout the plant body, but what is the expression pattern of BRAK, e.g. Promoter-GUS?

Thank you for making this point. Given the long time required for conducting Promoter-GUS assays, we chose qRT-PCR, a faster approach to reflect *BRAK*'s expression pattern. As shown in **Appendix Figure S1B**, *BRAK* gene were expressed throughout the plant, including roots, stems, leaves, flowers, fruits and seeds.

Q3: Although BRAK is a gene discovered in tomato, the Arabidopsis homologue AT1G74360 should be mentioned in the discussion. AT1G74360 has been shown in several papers to be involved in immune response.

Thank you for making this point. As suggested, we have discussed the role of Arabidopsis homologue AT1G74360 in immunity in **Line 299-301**.

Response to Referee #2:

The authors have identified a class-IV LRR-RLK, BRAK, which is induced upon infection of tomato with *B. cinerea*. This protein appears directly involved in the defense response by making plants more resistant. Interestingly, BRAK also positively influences plant growth. BRAK was also found by Y2H to interact with PSKR1, an RLK known from Arabidopsis to bind PSK, a signal triggered during defense responses. Interestingly, als PSKR1 has a positive effect on defense responses and plant growth. The two RLKs seem to mutually activate each other by phosphorylation at Tyr and Thr residues and these phospho-sites are essential for the positive effect of the RLKs on defense and growth control.

The manuscript is very interesting in that it describes a positive effect of RLKs on defense and growth - which is rather uncommon. It also describes the interaction of RLKs and their effect on phosphorylation. Since there are a number of RLK-RLK interactions reported and described in detail (e.g. FER, BRI1, FLS2, BAK1....), the finding of an additional RLK-RLK interaction is not a major break-through. However, if these RLKs indeed boost both resistance to pathogens and growth, this would be a very interesting entry point for the improvement of crop production.

Hence, to make this manuscript indeed a major step forward, the authors should show that their tomato plants indeed perform better / are more resistant and give at least a similar yield as the control plants. While the authors describe positive effects on seedling growth, they fail to quantify this supposed growth benefit at adult stages or in terms of fruit production.

Thank you for your valuable comments. To address this concern, we have conducted thorough assessments of plant phenotype during the reproductive stage, as well as measure fresh weight per fruit and fruit number per plant for WT, mutants and over-expression lines. Briefly, the fruit number, fresh weight per fruit and total fruit yield were significantly increased in both OE-BRAK and OE-PSKR1 plants, while decreased in *brak* and *pskr1* mutants. The new data are shown in **Figure 2G-K and EV1**, the text of the manuscript has been revised accordingly (**Line 150-153, Line 200-201**).

Other comments:

Q1: Fig. 1B: the expression is relative to what?

Thank you for making this point. The expression level in **Figure 1B** is relative to the mock condition. We have clarified this in the figure legend.

Q2: Fig. 1C: they should provide a co-localization with a known plasma membrane-localized protein.

Thank you for your suggestion. The well-known plasma membrane-localized protein, FLS2-mCherry has now been co-transfected with BRAK-GFP in **Figure 1C**.

Q3: Lane 150: where is the connection between quantum yield and defense/growth? Please explain to the non-specialized reader.

Thank you for your suggestion. The connection between quantum yield and defense/growth has been clarified in **Line 427-430**. Specifically, when plants face threats like pests or diseases, they often need to divert resources away from growth processes toward defense mechanisms. This reallocation of resources can affect the quantum yield because less energy is being used for growth and more for defense. Consequently, the decrease in energy allocation to growth leads to a reduction in quantum yield (Rohacek et al., 2008).

Q4: Lane 161 ff: of all the clones that were identified, why did they specifically choose PSKR1? Could all the others not be confirmed?

Thank you for making this point. The focus was on the interaction between LRR-RLKs. We identified two LRR-RLKs in total, one being Solyc04g064940 which belongs to VII-a, showing no significant response to *B. cinerea*. The other one was PSKR1, which exhibits similar phenotypes to BRAK. Therefore, we chose PSKR1 for further study. We have updated this in the revised manuscript (**Line 169-172**).

Q5: Lane 189: why was gene silencing used here? Is it that the CRISPR mutant in *brak* would not show the same phenotype? Or was it practical reasons?

Thank you for your question. Initially, we crossed OE-*PSKR1* with the *brak* mutant,

However, we failed to obtain the seeds from this cross. As an alternative approach, we used VIGS technology to silence the *BRAK* gene in OE-*PSKR1* plants. Additionally, we silenced the *BRAK* gene in WT plants as a control. Therefore, we did not use the *brak* mutant, not because it would not show the same phenotype, but rather for practical reasons.

Q6: Lane 193 and elsewhere: while the authors provide convincing evidence that PSKR1 and BRAK interact and are both important for PSK-induced responses, they still repeatedly state that BRAK functions downstream of PSK. What lets them come to this conclusion? Please explain.

Thank you for your question. We apologize for any confusion. When we say that BRAK functions downstream of PSK, it means that BRAK is a component of the PSK signaling pathway. Upon perceiving the PSK signal, BRAK then acts upon it, which then plays a role in mediating downstream responses. Therefore, we state that BRAK functions downstream of PSK. We have clarified this in manuscript (**Line 371-374**).

Q7: Lane 212 and further down: since the phosphorylation of BRAK is important for its function (and the same for PSKR1), is the autophosphorylation of these proteins influenced/triggered by the interacting RLK? In other words, is autophosphorylation of PSKR1 equally effective when incubating with a kinase-dead BRAK? And vice versa, of course. That would provide additional insight into the dynamics of this protein-protein interaction that should be technically feasible.

Thank you for your valuable suggestion. we have co-incubated purified BRAKJK and kinase-dead PSKR1JK proteins *in vitro* and compared the autophosphorylation levels of BRAK with or without kinase-dead PSKR1JK. Similarly, the effect of BRAK on the autophosphorylation of PSKR1 were also investigated. As shown in new data of **Figure EV2D-E**, we find that the autophosphorylation of these two RLK was not influenced by this interaction. The text of the manuscript has been revised accordingly (**Line 232-236**).

Q8: Fig. 8: in their RNA seq experiment, could the authors confirm the findings of the study by Courbier et al (which their RLK-identification was based

on)?

Thank you for making this point. We evaluated the gene expression of clade IV LRR-RLKs based on our transcriptomic data (**Table EV4**), as shown in the figure below. Similar to the findings reported by Courbier et al. (Plant Physiol, 2021), no genes in clade IV were downregulated following *B. cinerea* infection in wild-type plants. Furthermore, the genes identified as upregulated in Courbier's study exhibited similar degrees of upregulation, with BRAK showing the most significant increase. These findings corroborate well with the results reported by Courbier et al..

The transcript abundance of tomato LRR-RLK subfamily IV genes in leaves at 24 hours after *B. cinerea* infection. The transcript abundance under mock condition was defined as 1. Data

are from our RNA Seq (**Table EV4**) .

Q9: Lane 337: the phosphorylation of BRAK and PSRK1 are clearly not essential for plant growth. In fact, both genes are dispensable since the double mutant seems to grow fine. It's the fine regulation that is changed.

You are correct. We have revised the statement accordingly in **Line 357-358**.

Minor comments:

Q1: The abstract should mention the plant species that this work is about. Also spell out MST.

Thank you for your suggestion. The plant species and Microscale Thermophoresis (MST) have been added into the abstract.

Q2: In general, there is an inconsistent use of past and present tense. Description

of experiments should be in past tense, statements on facts/established knowledge or interpretations, however, should be in present tense (since they hold true not only in the past but also now). This tremendously facilitates the reading.

Thank you for making this point. We have corrected the tense in the revised manuscript.

Q3: Lane 47: progresses should probably be processes.

We are sorry. We have corrected this in the revised manuscript.

Q4: Lane 56: kinase activity is not necessary for all signaling activities of RLK.

Thank you for pointing that out. We have corrected it in the **Line 55-58**.

Q5: Lane 73 and elsewhere: the plant species is not always clear. Here, it is BRI1 of barley? Or is the growth rate in Arabidopsis barely reduced?

We apologize for any confusion. The experiments mentioned are conducted in barley. We have clarified this in **Line 73** and reviewed the entire text for accuracy.

Q6: Lane 74: What are CLVs? Explain.

Thank you for pointing that out. CLVs refer to the CLAVATs, which is essential for regulating the development and maintenance of plant stem cells, as mentioned in **Line 75-76**.

Q7: Lane 93-96: this is not a correct sentence (grammar-wise).

Thank you for pointing that out. We have rewritten the sentence in **Line 94-97**.

Q8: Fig. 2G, 6D, and 7E: label the panels to make the interpretation easier. It is not clear whether the labelling of the panel below is also ment for the one above.

Thank you for your suggestion. We have labeled the panels in **Figures 2G, 6D, and 7E** to improve clarity.

Q9: Fig. 5E and elsewhere: it is surprising to see such tiny SD with only 3 samples tested. Are the error bars perhaps SEM?

Thank you for pointing that out. We apologize for the mistake. Indeed, the standard deviation (SD) was calculated based on 8 samples, not 3. We have made the corrections in the figure legends.

Dear Dr. Shi,

Thank you submitting a revised version of your manuscript. It was sent to the same reviewers that originally appraised your work; their comments are attached to the bottom of this email. As you will see, both are satisfied with the changes you made. I would, though, like to consider incorporating the final small changes suggested by Reviewer #2. Before we can move forwards towards publication of your manuscript, there are some remaining editorial points which need to be addressed. In this regard, would you please:

- acknowledge funding in our online submission system from the National Key Research and Development Program of China (2023YFD2300701), the National Natural Science Foundation of China (32172650, 32302639), the Starry Night Science Fund of Zhejiang University Shanghai Institute for Advanced Study (SN-ZJU-SIAS-0011) and China Postdoctoral Science Foundation (2022M722800, 2023T160572),
- select five keywords,
- change the title of the 'Conflict of Interests' statement to the 'Disclosure and Competing Interests Statement',
- remove the author credit section from the manuscript,
- complete the reporting section on the author checklist,
- upload Figure 2 as a single figure file,
- upload tables EV1-EV7 individually,
- include a Reagents and Tools table,
- provide specific URLs for PRJNA1041583, PRJNA1144371 datasets in the data availability statement,
- correct the figure legend for figure 5e-f, where the error bar definition is mislabeled as figure 5d-e,
- correct the figure legend for figures 6c, e; 7d, f, where the "n" related information is mislabeled as 6e, f; 7f, g,
- define the annotated p values $^*/a/b/c/d/ab/bc/cd/abc$ as well as provide the exact p-values for the same in the legend of figure 1b; 2d, f, i-m, n, o; 4b, e-f; 5b, e-f; 6e, g; 7f, h; 8d; EV 1d-e; as appropriate,
- define all box plots in terms of minima, maxima, centre, bounds of box and whiskers, and percentile in the legends of figures 2k; EV 1e,
- check author email shutingding@zju.edu.cn, which is not functional, and
- remove the "Supplemental Data" section from the manuscript.

We include a synopsis of the paper (see <http://emboj.embopress.org/>). Please provide me with a two-sentence general summary statement and 3-5 bullet points that capture the key findings of the paper.

We also need a summary figure for the synopsis. The size should be 550 wide by [200-400] high (pixels). You can also use something from the figures if that is easier.

I look forward to receiving these changes. EMBO Press is an editorially independent publishing platform for the development of EMBO scientific publications.

Best wishes,

William

William Teale, PhD
Editor
The EMBO Journal
w.teale@embojournal.org

- a point-by-point response to the referees' comments, with a detailed description of the changes made (as a word file).
- a word file of the manuscript text.
- individual production quality figure files (one file per figure)

- a complete author checklist, which you can download from our author guidelines (<https://www.embopress.org/page/journal/14602075/authorguide>).

- Expanded View files (replacing Supplementary Information)

We realize that it is difficult to revise to a specific deadline. In the interest of protecting the conceptual advance provided by the work, we recommend a revision within 3 months (12th Dec 2024). Please discuss the revision progress ahead of this time with the editor if you require more time to complete the revisions. Use the link below to submit your revision:

Referee #1:

I have verified that the authors have appropriately addressed the referee's comments. I think the paper deserves to be accepted.

Referee #2:

The manuscript by Ding et al is a revised version describing the effect of BRAK and PSKR on plant growth and immune responses, which, rather unexpectedly, are both improved. The authors have addressed the points raised in my first review. I have a few points that should still be addressed, most being minor:

In the answer to my question Nr. 5, the authors state that PSKR_{oE} lines could not be obtained in the brak mutant background, which is why they used silencing of BRAK. I'm a bit surprised that the authors don't mention this point in their manuscript. After all, this might hint at an important, possibly mutual, regulatory function of PSKR and BRAK. In the absence of BRAK, elevated levels of PSKR have deleterious effects. I don't ask the authors to follow this up in the context of this manuscript, but I believe it is worth mentioning this point, and I didn't find it in the manuscript.

Lane 232-236: in the new text, the kinase-dead versions of BRAK and PSKR should be labelled accordingly ("KM") as introduced on lane 222. Otherwise, the nomenclature is inconsistent.

Q8: is the confirmation of the published data by the results presented here specifically mentioned somewhere? This is useful information.

Lane 355 and elsewhere: statements on conclusions have to be in present tense since they still hold true. Here: Our data show that BRAK and PSKR phosphorylate each other..... promotes their interaction and is crucial....

Dear Dr. William Teale and Referees,

We are pleased to resubmit our revised manuscript entitled “A novel LRR-RLK BRAK reciprocally phosphorylates PSKR1 to enhance growth and defense in tomato” (EMBOJ-2024-117048R1) by Ding et al., for publication in the *EMBO Journal*. We appreciate the constructive comments and suggestions made by the Editors and the Reviewers that have helped us tremendously in improving our manuscript.

We have addressed all the concerns raised by the Editors and the Reviewers. The changes incorporated into the original manuscript have been highlighted in red color in the revised manuscript. The text of the manuscript has also been checked carefully, and we have been careful to adhere to the format of the *EMBO Journal*. Our point-by-point responses to the Editors and the Referees comments are detailed below.

We believe that our revisions address all the points raised and that the revised version is acceptable for publication in the *EMBO Journal*. Please do not hesitate to contact us if further changes to the manuscript are required.

Sincerely,

Kai Shi, Ph. D

Department of Horticulture
Zhejiang University, China.
E-mail: kaishi@zju.edu.cn

October 3, 2024

Response to Editor:

Thank you submitting a revised version of your manuscript. It was sent to the same reviewers that originally appraised your work; their comments are attached to the bottom of this email. As you will see, both are satisfied with the changes you made. I would, though, like to consider incorporating the final small changes suggested by Reviewer #2. Before we can move forwards towards publication of your manuscript, there are some remaining editorial points which need to be addressed. In this regard, would you please:

- 1. acknowledge funding in our online submission system from the National Key Research and Development Program of China (2023YFD2300701), the National Natural Science Foundation of China (32172650, 32302639), the Starry Night Science Fund of Zhejiang University Shanghai Institute for Advanced Study (SN-ZJU-SIAS-0011) and China Postdoctoral Science Foundation (2022M722800, 2023T160572),**

Funding has been acknowledged in online submission system.

- 2. select five keywords,**

Five keywords have been selected.

- 3. change the title of the 'Conflict of Interests' statement to the 'Disclosure and Competing Interests Statement',**

The "Conflict of interest" section has been renamed to "Disclosure and Competing Interests Statement"

- 4. remove the author credit section from the manuscript,**

The Author Contributions section has been removed.

- 5. complete the reporting section on the author checklist,**

The reporting section on the author checklist has been completed.

- 6. upload Figure 2 as a single figure file,**

Figure 2 were uploaded as a single figure file.

7. include a Reagents and Tools table,

Reagents and Tools table have been uploaded.

8. upload tables EV1-EV7 individually,

The Tables were uploaded as suggested.

9. provide specific URLs for PRJNA1041583, PRJNA1144371 datasets in the data availability statement,

We have added specific URLs for PRJNA1041583, PRJNA1144371 datasets in Line 566-567.

10. correct the figure legend for figure 5e-f, where the error bar definition is mislabeled as figure 5d-e,

We are sorry and have corrected the mistake.

11. correct the figure legend for figures 6c, e; 7d, f, where the "n" related information is mislabeled as 6e, f; 7f, g,

We are sorry and have corrected the mistake.

12. define the annotated p values $*/a/b/c/d/ab/bc/cd/abc$ as well as provide the exact p-values for the same in the legend of figure 1b; 2d, f, i-m, n, o; 4b, e-f; 5b, e-f; 6e, g; 7f, h; 8d; EV 1d-e; as appropriate,

We have defined p values in the figure legends.

13. define all box plots in terms of minima, maxima, centre, bounds of box and whiskers, and percentile in the legends of figures 2k; EV 1e,

We have defined all box plots in the legends of figures 2k; EV 1e.

14. check author email shutingding@zju.edu.cn, which is not functional,

We are sorry and have corrected the mistake.

15. remove the "Supplemental Data" section from the manuscript.

The "Supplemental Data" section has been removed.

- 16. We include a synopsis of the paper (see <http://emboj.embopress.org/>). Please provide me with a two-sentence general summary statement and 3-5 bullet points that capture the key findings of the paper.**

The synopsis of the paper has been uploaded as suggested.

- 17. We also need a summary figure for the synopsis. The size should be 550 wide by [200-400] high (pixels). You can also use something from the figures if that is easier.**

The summary figure has been uploaded according to your suggestion.

Response to Referee #1:

I have verified that the authors have appropriately addressed the referee's comments. I think the paper deserves to be accepted.

Thank you.

Response to Referee #2:

The manuscript by Ding et al is a revised version describing the effect of BRAK and PSKR on plant growth and immune responses, which, rather unexpectedly, are both improved. The authors have addressed the points raised in my first review. I have a few points that should still be addressed, most being minor:

In the answer to my question Nr. 5, the authors state that PSKR_OE lines could not be obtained in the brak mutant background, which is why they used silencing of BRAK. I'm a bit surprised that the authors don't mention this point in their manuscript. After all, this might hint at an important, possibly mutual, regulatory function of PSKR and BRAK. In the absence of BRAK, elevated levels of PSKR have deleterious effects. I don't ask the authors to follow this up in the context of this manuscript, but I believe it is worth mentioning this point, and I didn't find it in the manuscript.

Thank you for your suggestion. We have mentioned this point in the revised version of our manuscript in **Line 196-197**.

Lane 232-236: in the new text, the kinase-dead versions of BRAK and PSKR should be labelled accordingly ("KM") as introduced on lane 222. Otherwise, the nomenclature is inconsistent.

Thank you for pointing that out. We have corrected it in the **Line 235**.

Q8: is the confirmation of the published data by the results presented here specifically mentioned somewhere? This is useful information.

Thank you for your suggestion. We have mentioned it in the revised manuscript (**Line 297-298, Appendix Figure S4**).

Lane 355 and elsewhere: statements on conclusions have to be in present tense since they still hold true. Here: Our data show that BRAK and PSKR phosphorylate each other..... promotes their interaction and is crucial...

Thank you for making this point. We have corrected the tense in the revised manuscript in **Line 317, 356, 358, and 380**.

Dear Kai,

I am pleased to inform you that your manuscript has been accepted for publication in the EMBO Journal.

Congratulations on completing a really nice project!

Best wishes,

William

William Teale, PhD
Editor
The EMBO Journal
w.teale@embojournal.org
